**METHOD**

# aMeta: an accurate and memory-efficient ancient metagenomic profiling workflow

Zoé Pochon[1,2†], Nora Bergfeldt[1,3,4†], Emrah Kırdök[5], Mário Vicente[1,2], Thijessen Naidoo[1,2,6,7], Tom van der Valk[1,4], N. Ezgi Altınışık[8], Maja Krzewińska[1,2], Love Dalén[1,3], Anders Götherström[1,2†], Claudio Mirabello[9†], Per Unneberg[10†] and Nikolay Oskolkov[11*†] [ID]

†Zoé Pochon, Nora Bergfeldt, Anders Götherström, Claudio Mirabello, Per Unneberg, and Nikolay Oskolkov shared authorship.

*Correspondence: Nikolay.Oskolkov@biol.lu.se

[11] Department of Biology, Science for Life Laboratory, National Bioinformatics Infrastructure Sweden, Lund University, Lund, Sweden Full list of author information is available at the end of the article

## Abstract

Analysis of microbial data from archaeological samples is a growing field with great potential for understanding ancient environments, lifestyles, and diseases. However, high error rates have been a challenge in ancient metagenomics, and the availability of computational frameworks that meet the demands of the field is limited. Here, we propose aMeta, an accurate metagenomic profiling workflow for ancient DNA designed to minimize the amount of false discoveries and computer memory requirements. Using simulated data, we benchmark aMeta against a current state-of-the-art workflow and demonstrate its superiority in microbial detection and authentication, as well as substantially lower usage of computer memory.

**Keywords:** Ancient metagenomics, Pathogen detection, Microbiome profiling, Ancient DNA

## Background

Historically, ancient DNA (aDNA) studies have focused on human and faunal evolution and demography, extracting and analyzing predominantly eukaryotic aDNA [1–3]. With the development of next-generation sequencing (NGS) technologies, it was demonstrated that host-associated microbial aDNA from eukaryotic remains, which was previously treated as a sequencing by-product, can provide valuable information about ancient pandemics, lifestyle, and population migrations in the past [4–6]. Modern technologies have made it possible to study not only ancient microbiomes populating eukaryotic hosts, but also sedimentary ancient DNA (sedaDNA), which has rapidly become an independent branch of palaeogenetics, delivering unprecedented information about hominin and animal evolution without the need to analyze historical bones and teeth [7–12]. Previously available in microbial ecology, meta-barcoding methods lack validation and authentication power, and therefore, shotgun metagenomics has become the *de facto* standard in ancient microbiome research [13]. However, accurate detection,

abundance quantification, and authentication analysis of microbial organisms in ancient metagenomic samples remain challenging [14]. This is due to the limited amount of microbial aDNA and the exceptional variety of both host-associated and invasive microbial communities that have been populating ancient samples when living and *post-mortem*. In particular, the presence of modern contamination can introduce biases in the analysis of aDNA data. All of these technical and biological factors can lead to a high rate of false-positive and false-negative microbial identifications in ancient metagenomic studies [15].

When screening for the presence of microbial organisms with available reference genomes, we aim to assign a taxonomic label to each aDNA sequence. For this purpose, there are two dominant approaches: composition, aka *k*-mer taxonomic classification, and alignment-based methods. For the former, the Kraken family of tools [16, 17] is one of the most popular in ancient metagenomics, while for the latter, general purpose aligners such as BWA [18] and Bowtie2 [19], but also aligners specifically designed for the analysis of metagenomic data, such as MALT [20], are among the most commonly used.

Unlike the alignment approach, where each aDNA sequence is positioned along the reference genome based on its similarity to it, the *k*-mer taxonomic classification uses a lookup database containing *k*-mers and Lowest Common Ancestor (LCA) information for all organisms with available reference genomes. DNA sequences are classified by searching the database for each *k*-mer in a sequence and then by using the LCA information to determine the most specific taxonomic level for the sequence. Advantages of the classification-based approach are high speed and a wide range of candidates (database size), while disadvantages include difficulty in validation and authentication which can often lead to a high error rate in the classification-based approach. In contrast, the alignment-based approach with, for example, MALT provides more means of validation and authentication, while being relatively slow, more resource-demanding, and heavily dependent on the selection of reference sequences included in the database. Technical limitations such as computer memory (RAM) often hinder the inclusion of a large amount of reference sequences into the database, which might result in a high false-negative rate of microbial detection. In practice, due to the very different nature of the analyses and reference databases used, the outputs from classification and alignment approaches often contradict each other, bringing additional confusion to the ancient metagenomics research community. In fact, both approaches have strengths but also profound weaknesses that can lead to substantial analysis error, if not properly taken into account.

Here, we define two types of errors common to ancient metagenomics: (1) the detection error and (2) the authentication error. The detection error comes from the difficulty to correctly identify microbial presence or absence irrespective of ancient status. This can happen due to many reasons such as overly relaxed or too conservative filtering. This error is not specific to ancient metagenomics but represents a general challenge that is also valid for the field of modern metagenomics. In contrast, the authentication error in our case is mainly related to the ancient status of detected organisms and caused by modern contamination that is typically present in archaeological samples. Often, inaccurate data processing and handling can lead to the erroneous discovery of a modern contaminant as being of ancient origin, and vice versa, of an ancient microbe

as being modern. Therefore, the major goals of an ancient microbiome reconstruction are to establish accurate evidence that a microbe (a) truly exists in a sample and (b) is of ancient origin.

Here, we aim to combine the strengths of both classification- and alignment-based approaches to develop an ancient metagenomic profiling workflow, aMeta, with low detection and authentication errors. For this purpose, we use KrakenUniq [21, 22] — which is suitable for working in low-memory computational environments — for initial taxonomic profiling of metagenomic samples and informing MALT reference database construction, followed by LCA-based MALT alignments, and a comprehensive validation and authentication analysis based on the alignments. We report that a KrakenUniq-based selection of microbial candidates for inclusion in the MALT database dramatically reduces resource usage of aMeta compared to metagenomic profiling with MALT alone. We evaluated our workflow using simulated ancient metagenomic data and benchmarked it against Heuristic Operations for Pathogen Screening (HOPS) [23], which is probably the most popular and *de facto* standard ancient metagenomic pipeline currently. We demonstrate that due to its additional breadth/evenness of coverage filtering, greater database size, and flexible authentication scoring system, the combination of KrakenUniq and MALT implemented in our workflow results in a higher sensitivity vs. specificity balance for the detection and authentication of ancient microbes compared to HOPS, given identical computer memory available. Importantly, aMeta consumed nearly half as much computer memory as HOPS on a benchmark simulated ancient metagenomic dataset.

## Results

The aMeta workflow overview is shown in Fig. 1. It represents an end-to-end processing and analysis framework implemented in Snakemake [24] that accepts raw sequencing data as a set of files, usually belonging to a common project, and outputs a ranked list of

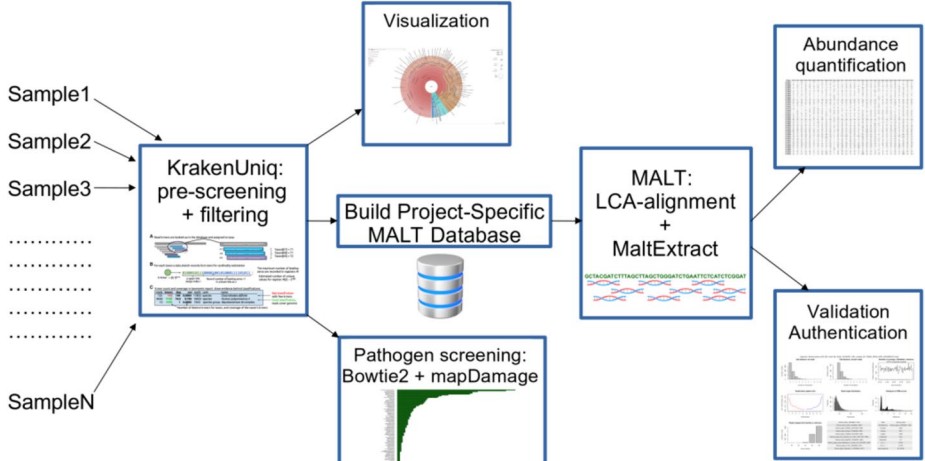

**Fig. 1** aMeta: ancient metagenomic profiling workflow overview. The workflow represents a combination of taxonomic classification + filtering steps with KrakenUniq that allows to establish a list of microbial candidates for further dynamic building of a MALT database, running LCA-based alignments with MALT against the database, and performing validation + authentication analysis based on the alignments

detected ancient microbial species together with their abundances for each sample, as well as a number of validation and authentication plots for each identified microorganism in each sample. In other words, the workflow leverages a convenient high-level summary of several authentication and validation metrics that evaluate detected microbes based on the evidence of their presence and ancient status.

Here, we provide a detailed description of each step implemented in aMeta. The workflow accepts raw metagenomic data in a standard *fastq* format containing sequenced DNA reads, removes sequencing adapters with Cutadapt [25], and selects reads of length above 31 bp which have a good taxonomic specificity. Next, the workflow runs KrakenUniq [21, 22] (we refer to this step as "pre-screening"), a fast and accurate *k*-mer-based tool which is capable of operating in low-memory computational environments [22]. KrakenUniq performs a taxonomic classification of the aDNA sequences and reports a number of *k*-mers unique within each taxon's reference genome, which can be considered as a good approximation to the breadth of coverage information; see Additional file [1]: Fig. S1. Indeed, a greater number of unique *k*-mers implies a broader breadth of coverage since reads span more unique regions, distribute across the reference genome, and encompass more base pairs of the reference. The number of unique *k*-mers is an essential filter of aMeta which significantly improves its accuracy (default: 1000 unique *k*-mers, can be configured by the user). Generally, breadth of coverage information is obtained through alignments; therefore, the advantage of KrakenUniq is that it is capable of providing an estimate of the breadth of coverage via *k*-mer-based read classification without the need for explicit alignments.

Figure [2] schematically demonstrates why detection of microbial organisms based solely on *depth of coverage*, sometimes referred to as "coverage" in the literature, might lead to false-positive identifications. Depth of coverage is equivalent to the total number of mapped reads, normalized by the length of the reference genome. Suppose we have a toy example with a reference genome of length $4 * L$ and 4 reads of length $L$ mapping to the reference genome. When a microbe is truly detected, the reads should map *evenly*, which means they should be distributed randomly and hence in a relatively even manner

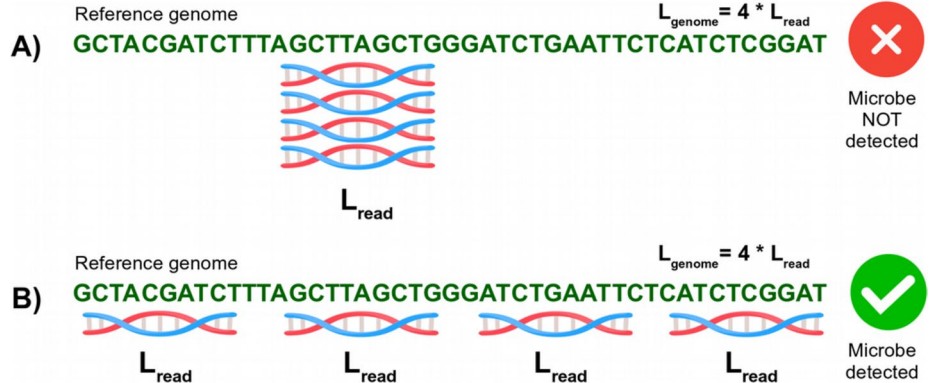

**Fig. 2** Schematic demonstration of the difference between depth and breadth/evenness of coverage concepts. Two read alignment scenarios, **A** and **B**, have an identical depth of coverage of $N_{reads} * L_{read} / L_{genome} = 4 * L_{read} / 4 * L_{read} = 1X$. However, the reads are spread unevenly in case **A** and evenly in case **B**. The latter has a higher breadth of coverage (100% in contrast to 25% for the former) and evenness of coverage. Scenario **B** corresponds to a true-positive hit, while scenario **A** is typical of a false-positive microbial detection

across the reference genome; see Fig. 2B. In this case, the mapped reads have satisfactory *breadth of coverage* (fraction of covered reference nucleotides) and *evenness of coverage* (uniformness or consistency with which sequenced reads are distributed across the reference genome). In the toy example in Fig. 2B, all 4 reads map at unique positions and such alignments provide a perfect breadth of coverage of 100%. In contrast, in case of misaligned reads, where reads originating from species A are incorrectly mapped to the reference genome of species B, it is common to observe read stacking in regions of high sequence conservation, which is the case in Fig. 2A where all 4 reads align at the same position (see also Additional file 1, Fig. S2 for a real data example, where reads from unknown microbial organisms are deliberately forced to map exclusively to the *Yersinia pestis* reference genome). In the toy example in Fig. 2A, the read alignments have a rather poor breadth of coverage of 25%. Note that the non-overlapping reads in Fig. 2B will likely cover more unique *k*-mers (providing large enough *k*) compared to the stacked reads in Fig. 2A. Therefore, we consider the breadth of coverage, which is conceptually related to the number of unique *k*-mers delivered by KrakenUniq, to be of crucial importance for robust filtering in our workflow.

In addition to the breadth of coverage filtering, low-abundance microbes are removed in aMeta based on their depth of coverage, which is estimated by the number of reads assigned to each taxon (default: 200 reads, which can be adapted by the user). Filtering by the depth of coverage is also important for subsequent validation and authentication steps, as some of these may not be statistically robust enough when performed on low-abundant microbes. Therefore, aMeta uses a rather conservative approach and focuses on reasonably abundant species with even coverage which are more likely to actually be present in the samples (Fig. 3).

For pre-screening with KrakenUniq, we built two different databases of reference sequences: (1) a complete NCBI non-redundant NT database (referred to as full NCBI NT), currently used by default in BLASTN [26], that included all eukaryotic and prokaryotic genomes available at NCBI, December 2020; (2) a microbial version of the full NCBI NT database (referred to as Microbial NCBI NT), consisting of all microbial genomic sequences (bacteria, viruses, archaea, fungi, protozoa and parasitic worms) as well as the human genome and complete eukaryotic genomes from NCBI. The former database can be used for comprehensive screening of both eukaryotic (mammals, plants, etc.) and microbial organisms, while the latter is more than half the size, and is sufficient for microbial profiling only. Both databases are publicly available to the wider scientific community through the SciLifeLab Figshare at https://doi.org/10.17044/scilifelab.20205504 and https://doi.org/10.17044/scilifelab.20518251.

When comparing different KrakenUniq databases, we found that database size played an important role in robust microbial identification. Specifically, small databases tended to have higher false-positive and false-negative rates for two reasons. First, microbes present in a sample whose reference genomes were not included in the KrakenUniq database could obviously not be identified, hence the high rate of false negatives of smaller databases. Second, microbes in the database that were genetically similar to the ones in a sample appeared to be more often erroneously identified, which contributed to the high rate of false positives of smaller databases. For more details, see the sub-section "Effect of database size" of the "Results" section.

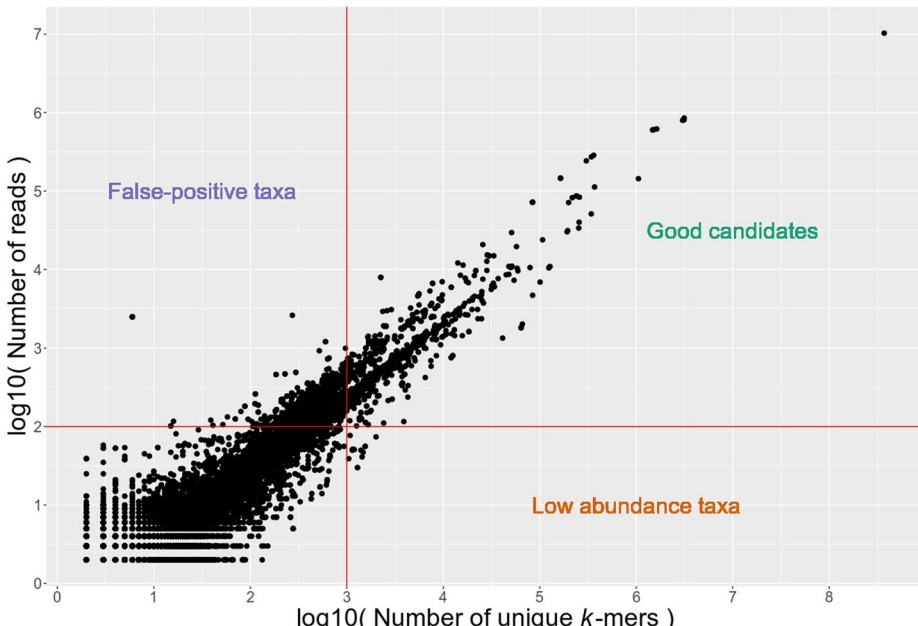

**Fig. 3** Depth (number of reads specific to a taxon) and breadth (number of unique *k*-mers) of coverage reported by KrakenUniq for microbial taxa in a metagenomic sample. Taxa with large amounts of unevenly mapped reads, and hence low breadth of coverage, are considered to be false-positive identifications (left upper corner). Red solid horizontal and vertical lines mark optional depth (~ 100 reads) and breadth (~ 1000 unique *k*-mers) of coverage filters applied to the KrakenUniq output

Although the technique of filtering the KrakenUniq output by depth and breadth of coverage is reliable for establishing the presence of an organism in a sample, the findings of KrakenUniq have to be authenticated, i.e., their ancient status needs to be confirmed, which is impossible to do with the taxonomic classification approach alone. Indeed, even if KrakenUniq has a fairly low detection error (see the "Background" section), it cannot provide any information about the ancient status of the detected microbes. Furthermore, additional validation based on the alignment summary statistics such as evenness of coverage and edit distance can enhance the detection accuracy of aMeta beyond the primary hard thresholds of breadth and depth of coverage applied to KrakenUniq output.

To validate the results from the KrakenUniq pre-screening step and further eliminate potential false-positive microbial identifications, aMeta performs alignments with the MALT aligner [20]. The main advantage of MALT and motivation for us to use it in aMeta was that MALT is a metagenomic-specific aligner which applies the Lowest Common Ancestor (LCA) algorithm in contrast to other traditional genomic aligners such as BWA [18] and Bowtie2 [19]. The LCA algorithm is particularly important when working with heterogeneous metagenomic sequencing data. More specifically, when performing competitive mapping to multiple reference genomes, it is important to correctly handle the reads mapping with the same affinity to several references (multi-mapping reads). Traditional genomic aligners would disregard the multi-mapping reads as ambiguous and non-informative. In contrast, the LCA algorithm in MALT keeps the multi-mapping reads within the taxonomic tree of related organisms and assigns the reads to the lower ancestor node in the tree. For example, if a read maps with the same number of mismatches to two species, the read will be assigned to their common genus and kept for

downstream analysis. Alternatively, aMeta users can also select Bowtie2 for faster and more memory-efficient alignments; see Additional file 2: S1. Indeed, Bowtie2 may be preferred by users because MALT is very resource-demanding. However, since Bowtie2 lacks LCA handling of multi-mapping reads, MALT is more suitable for metagenomic analysis.

In practice, only reference databases of limited size can be afforded when performing analysis with MALT, which might potentially compromise the accuracy of microbial detection; see Additional file 2: S2 for more details. In consequence, we aim to link the unique capacity of KrakenUniq to work with large databases with the advantages of MALT for the validation of results via an LCA alignment. For this purpose, aMeta dynamically builds a project-specific MALT database, based on a filtered list of microbial species identified by KrakenUniq. In other words, the combination of microbes across the samples, remaining after depth and breadth of coverage filtering of the KrakenUniq outputs, is used to build a MALT database, which allows the running of LCA-based MALT alignments using realistic computational resources.

The analysis strategy applied in the aMeta workflow is two-step. First, we pre-screen and classify microbial organisms in aDNA samples with KrakenUniq against the full NT or microbial NT database, a step that can be performed virtually on any computer, even a laptop. Second, we validate the findings by performing MALT LCA-based alignments against a project-specific database comprising microbial species identified at the first step by KrakenUniq. This two-step strategy provides a good balance between sensitivity and specificity of both microbial detection and authentication in aDNA metagenomic samples without imposing a large computational resource burden. On the one hand, the KrakenUniq step optimizes the sensitivity of microbial detection by using a large database that would otherwise be likely technically impossible for MALT to handle. On the other hand, the MALT step optimizes the specificity of microbial detection and authentication by performing LCA-based alignments suitable for computing various quality metrics. Note that the two-step design of aMeta minimizes potential conflicts between classification (KrakenUniq) and alignment (MALT) approaches by ensuring consistent use of the reference database.

As previously emphasized, microbial organisms identified by KrakenUniq and MALT in metagenomic samples need to be checked for their ancient status, i.e., authentication analysis is needed in order to discriminate truly ancient organisms from modern contaminants. For authentication of microbial organisms found in metagenomic aDNA samples, we applied the MaltExtract tool [23] to the LCA-based alignments produced by MALT and computed the deamination pattern [27, 28], read length distribution, average nucleotide identity (ANI) via percent identity to the reference, and edit distance (amount of mismatches) [23] metrics. Next, the breadth and evenness of coverage of reads aligned to each microbial reference genome were generated using SAMtools [29]; see the "Methods" section and Additional file 2: S3. In addition, the workflow automatically extracts alignments and the corresponding reference genome sequence for each identified microbial organism in each sample, allowing users to visually inspect the alignments, e.g., in the Integrative Genomics Viewer (IGV) [30], which provides intuitive interpretation of the quality metrics reported by aMeta. Finally, histograms of postmortem damage scores (PMD) are computed using PMDtools [31], which features a unique option of

likelihood-based inference of ancient status with a single read resolution. All the mentioned quality metrics are complementary and serve for more informed decisions about the presence and ancient status of microorganisms in metagenomic samples. A typical graphical output from aMeta is demonstrated in Fig. 4, which summarizes authentication and validation information for *Yersinia pestis*, the pathogen responsible for the plague, previously reported for the Gökhem2 (Gok2) individual [5]; see also Additional file 1: Fig. S3 for the Gökhem4 (Gok4) individual [5].

In addition to the graphical summary of quality metrics, aMeta delivers a table of microbial abundances quantified from both *rma6-* and SAM-alignments available from MALT. The alignments in *rma6* format are quantified using the *rma2info* wrapper script from the MEGAN tool [32]; see Additional file 2: S7, while a custom *awk* script is used for quantifying microbial abundance from SAM-alignments. A disadvantage of *rma6*, which is a primary MALT alignment format, is that it cannot be easily handled by typical bioinformatics software such as SAMtools. However, we found that the alternative alignments in SAM format delivered by MALT lack LCA information and therefore are not optimal either, since they essentially resemble the Bowtie2 alignments. Nevertheless, we believe the two ways of abundance quantification are complementary to each other. The LCA-based quantification from the *rma6* output of MALT might underestimate the true per-species microbial abundance, since many short conserved aDNA sequences originating from a species are assigned to higher taxonomic levels, e.g., genus level, and thus do not contribute to the species abundance. In contrast, the LCA-unaware quantification from the SAM output of MALT seems to overestimate the true per-species

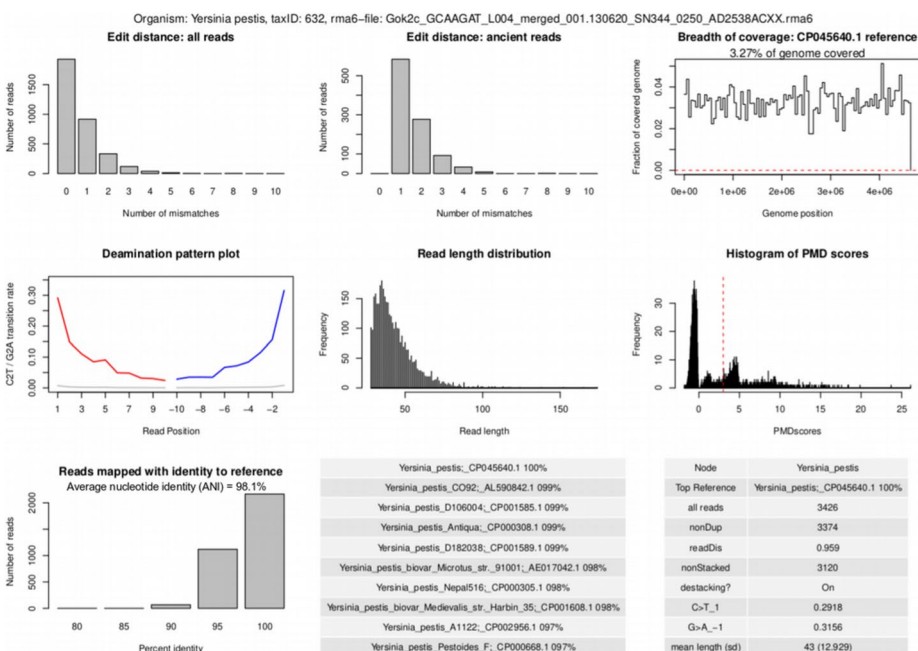

**Fig. 4** Authentication output of aMeta. Panels from left to right, top to bottom: **a** edit distance computed on all assigned reads, **b** edit distance computed on damaged reads, **c** evenness/breadth of coverage, **d** deamination pattern, **e** read length distribution, **f** PMD scores distribution, **g** number of reads assigned with an identity to a reference, **h** candidate reference sequences with percentages of mapped reads, and **i** MaltExtract statistics

microbial abundance since it counts absolutely all reads assigned to a species, including the non-specific multi-mapping reads, i.e., the ones that map with the same affinity to multiple homologous microbial organisms.

Within the aMeta workflow, we constructed and implemented a special authentication scoring system that should facilitate getting a quick user-friendly overview of potentially present ancient microbes; see "Methods" section and Additional file 2: S5 for more details. The score is computed per microbe per sample and represents a quantity that combines the eight validation and authentication metrics presented graphically in Fig. 4, more specifically (1) deamination profile, (2) evenness of coverage, (3) edit distance (amount of mismatches) for all reads, (4) edit distance (amount of mismatches) for damaged reads, (5) read length distribution, (6) PMD scores distribution, (7) number of assigned reads (depth of coverage), and (8) average nucleotide identity (ANI) via percent identity to the reference. The scoring system assigns heavier weights to evenness of coverage as an ultimate criterion for the true presence of a microbe, and deamination profile as the most important evidence of its ancient origin.

As one of the main outputs, aMeta delivers a heatmap summary of authentication scores for each detected microbe in each sample (Additional file 1: Fig. S4). The heatmap of scores, ranging from 0 (either not present or modern contaminant) to 10 (present and ancient), provides a quick and easy way for users to summarize the results of the ancient metagenomic analysis and make more informed decisions about hits to follow-up.

### Benchmarking aMeta on simulated data

We benchmarked aMeta against HOPS [23] which is one of the most widely used pipelines in the field of ancient metagenomics. Another popular general-purpose aDNA pipeline, nf-core/eager [33], implements HOPS as an ancient microbiome profiling module within the pipeline; therefore, we do not specifically compare our workflow with nf-core/eager but concentrate on differences between aMeta and HOPS in terms of computational resources and accuracy.

For robust comparison of the two approaches, we built a ground truth dataset which represents 10 ancient human metagenomic samples with various microbial compositions simulated with the gargammel tool [34]. To mimic potential contamination scenario, we simulated reads that were both host-associated (ancient) and contaminant (ancient and modern). We selected 35 microbial species that are commonly found across our aDNA projects [35, 36], and simulated their fragmented and damaged reads. In addition, Illumina adapters and sequencing errors were added to mimic typical ancient DNA raw genomic sequencing data; see the "Methods" section for details. To better resemble a typical situation in our ancient metagenomic studies [35, 36], we simulated bacterial reads of both modern and ancient origin. For example, when working with ancient dental calculus [35], one may often observe host-associated *Streptococcus pyogenes* or *Parvimonas micra*, which were simulated here as being of ancient origin. One can also find ancient exogenous bacteria of environmental origin, such as *Mycobacterium avium* and *Ralstonia solanacearum*, which were also simulated as ancient. In total, 18 out of 35 microbial species were simulated as ancient. We also added a number of modern bacterial contaminants, such as a few species from the *Burkholderia* and *Pseudomonas* genera, that are typically found on (blank) negative controls in our aDNA lab.

The contaminants were simulated with a moderate fragmentation level and no clear deamination or damage pattern. In total, 17 out of 35 microbial species were simulated as modern. In summary, the simulated ground truth dataset included both human and microbial DNA reads of ancient and modern origin, present at various ratios with varying levels of damage and fragmentation. We believe that this closely mimics a typical metagenomic composition scenario that we observe in various aDNA metagenomic projects [35, 36]; see the "Methods" section for more details.

Using this simulated ground truth dataset, we first aimed at comparing computer memory resources required by aMeta and HOPS. For this purpose, we ran aMeta on the simulated data using default settings and the Microbial NCBI NT database, which was the second largest among our pre-built databases; see the "Effect of database size" subsection for more details. For comparison, we also ran HOPS with default configuration parameters on our smallest database, the NCBI RefSeq database with complete microbial genomes, which however could only be used by HOPS on a computer node with at least 1 TB of RAM. In our computer resource benchmarking, we found that the design of aMeta (pre-screening with KrakenUniq followed by the dynamic construction of the MALT database) reduces computer memory load (RAM) by approximately two times compared to the resources required to accommodate the MALT database in the HOPS pipeline, as shown in Fig. 5. More specifically, aMeta used at most 353 GB of RAM on 20 threads, while HOPS required at most 685 GB of RAM on 1 thread, and 720 GB on 20 threads. This memory reduction in aMeta became possible due to two factors: (1) the recent low-memory development of KrakenUniq [22] and (2) the dynamic building of the MALT database after pre-screening with KrakenUniq. Moreover, the peak memory load of aMeta can be further reduced from 353 GB to approximately 140–150 GB

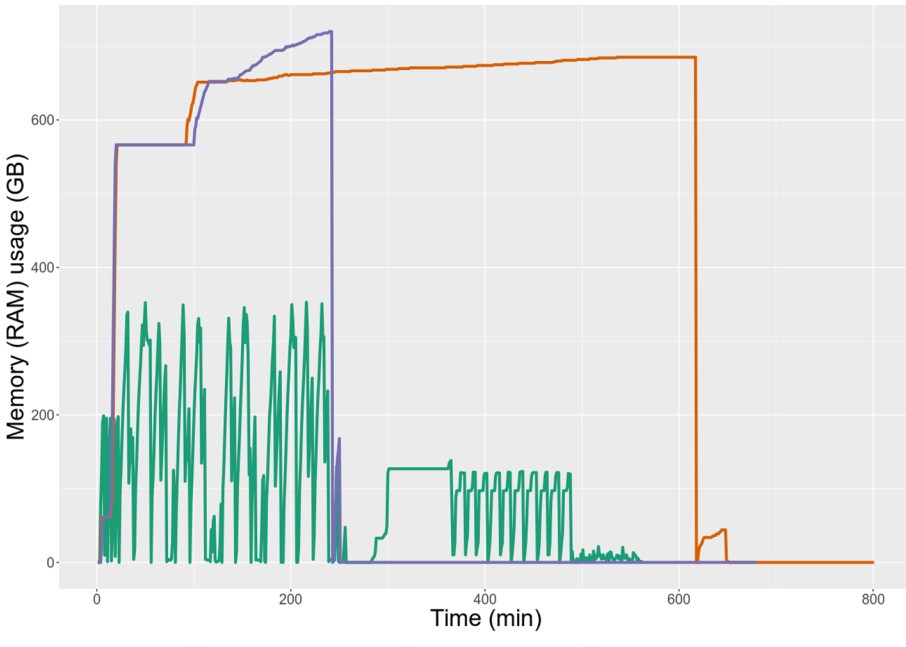

**Fig. 5** Comparison of aMeta vs. HOPS computer memory (RAM) usage. Peak memory load of aMeta on a benchmark dataset was approximately twice lower compared to peak memory load of HOPS

(a memory reduction of about 5 times compared to HOPS) for our benchmark dataset. This is because, when running aMeta, the irreducible memory consumption of MALT was only 138 GB, while the memory peak of 353 GB was observed on the KrakenUniq step, which in fact could be executed with even less available memory [22]. This, however, would lead to a longer computation time. Nevertheless, according to our tests (Additional file 1: Fig. S16), the new KrakenUniq development enables up to 10 times faster classification using a 450 GB reference database than with the previous versions, even on a computer cluster node with 128 GB of RAM, which was previously impossible without a node with at least 512 GB of RAM.

We conclude that aMeta is a more memory-efficient ancient metagenomic workflow compared to HOPS. However, one might be concerned that this superior computational efficiency can come at the price of reduced accuracy of metagenomic analysis, which would be undesirable. To address this, we computed microbial detection and authentication errors using the simulated ancient metagenomic dataset with the known ground truth described above.

We first sought to quantify the detection error of aMeta and HOPS, i.e., when a tool falsely reports the presence or absence of a microbe in a metagenomic sample, regardless of its ancient status. For this purpose, aMeta with default settings was run on the simulated dataset, and the microbial abundance matrix was computed by KrakenUniq after filtering for a breadth of coverage using the Microbial NCBI NT database. For comparison, HOPS with default configuration parameters was run using the complete microbial genomes NCBI RefSeq database, which was the largest database that was feasible to use for HOPS on a 1 TB of RAM computer cluster node. We quantified the abundance of microbial organisms detected by HOPS using MEGAN [32]. Next, both KrakenUniq and HOPS microbial abundance matrices were filtered using gradually increasing thresholds for the number of assigned reads, which is equivalent to filtering by depth of coverage. For each depth of coverage threshold applied to the abundance matrices, we compared microbial organisms identified by KrakenUniq and HOPS against the true list of organisms simulated by gargammel. As a criterion of overlap between the prediction and ground truth, we used two metrics: Intersection over Union (IoU), aka Jaccard similarity, and F1 score, which both quantify the balance between sensitivity and specificity of microbial detection by KrakenUniq and HOPS (Fig. 6). Illustrated by the solid lines in Fig. 6, it is demonstrated how Jaccard similarity and F1 score change at different depth of coverage thresholds applied to the KrakenUniq and HOPS microbial abundance matrices. The dashed horizontal line in Fig. 6 corresponds to the Jaccard similarity and F1 score computed using the depth and breadth of coverage thresholds set by default in aMeta, which can however be modified by the users. More specifically, by default, aMeta uses 1000 unique *k*-mers and 200 reads assigned to the taxon (taxReads) for filtering by depth and breadth of coverage, respectively. The default aMeta filtering thresholds were previously empirically determined from the analysis of over 1200 ancient metagenomic libraries [35, 36]. Nevertheless, the users are encouraged to experiment with the number of assigned reads threshold in the range of ~100–300 reads, and unique *k*-mers threshold in the range of ~500–1500 k-mers for their particular projects depending on sequencing depth and organism interest. As Fig. 6 shows, the default settings of aMeta result in nearly optimal Jaccard similarity and F1 score values obtained from filtering

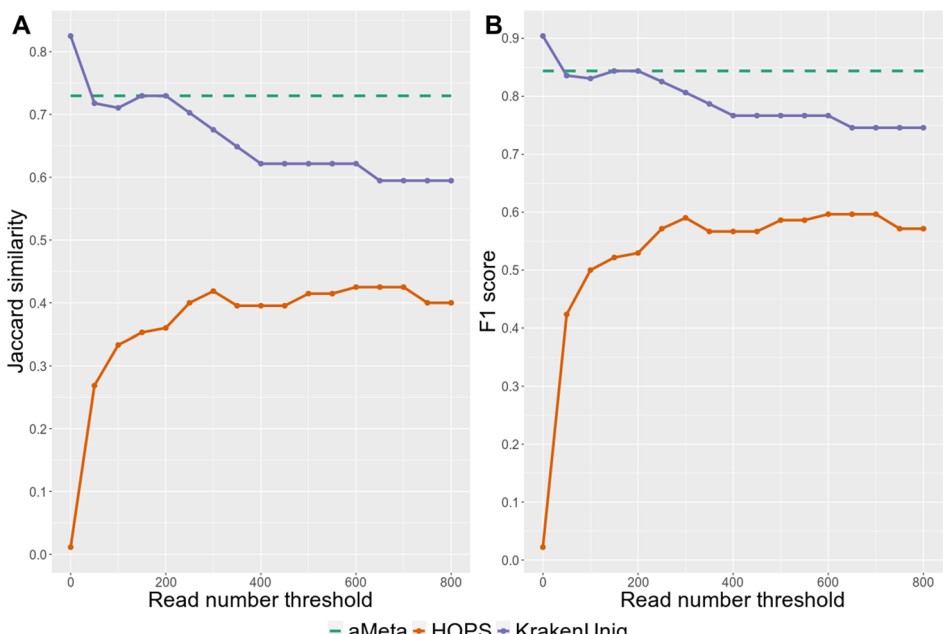

**Fig. 6** Microbial detection sensitivity vs. specificity comparison between KrakenUniq, HOPS, and aMeta (default settings), at different assigned reads thresholds: **A** Jaccard similarity and **B** F1 score, are computed with respect to the simulated microbial abundance ground truth

the KrakenUniq abundance matrix. Importantly, in Fig. 6, one can observe that irrespective of the depth of coverage threshold applied to the KrakenUniq and HOPS abundance matrices, the Jaccard similarity and F1 score for HOPS are always below the sensitivity vs. specificity level provided by KrakenUniq and aMeta. This conclusion is also confirmed by inspecting the accuracy of microbial composition reconstruction, as well as the numbers of false-positive and false-negative counts, at different read number thresholds (Additional file 1: Fig. S10).

The greater detection accuracy of aMeta is explained by two factors. First, since it is computationally feasible to use very large and phylogenetically diverse databases for taxonomic profiling with KrakenUniq and hence aMeta, this allows for the detection of microbial organisms that might be missed by HOPS due to their potential absence in the HOPS database (since it was not technically feasible to build and use as large and diverse HOPS database as it was possible for aMeta). Therefore, KrakenUniq and aMeta have a higher sensitivity for microbial detection. This conclusion is confirmed by Additional file 1: Fig. S5-S10, where the ground truth for the microbial presence-absence per sample is compared against the one reconstructed by aMeta and HOPS. For example, such simulated species as *Campylobacter rectus*, *Fusarium fujikuroi*, *Methylobacterium bullatum*, *Micromonas commoda*, *Micromonospora echinospora*, *Mycobacterium riyadhense*, *Nonomuraea gerenzanensis*, *Pseudomonas psychrophila*, and *Pseudomonas thivervalensis* were correctly identified by aMeta as present in the dataset, but not detected by HOPS in any simulated sample. Interestingly, *Campylobacter showae* was detected by HOPS instead of *Campylobacter rectus* because only the former was included in the HOPS database. This shows how a limited database size can impact not only the sensitivity (missed microbes) but also the specificity (falsely identified microbes) of microbial

detection. In total, HOPS missed 16 out of 35 simulated microbial species in all samples, while aMeta completely missed only 9 out of 35 microbes.

The second factor for increased accuracy of microbial detection by aMeta comes from the fact that, while the HOPS microbial abundance matrix can only be filtered by depth of coverage (*readDis* reported by HOPS is not used for filtering and cannot be considered as an optimal proxy for breadth of coverage), an additional breadth of coverage filter is available in KrakenUniq, and applied by aMeta, improving the robustness of microbial detection. Therefore, KrakenUniq and aMeta tend to have overall higher specificity for microbial detection. For example, such microbial species as *Mycobacterium avium*, *Nocardia brasiliensis*, *Rhodopseudomonas palustris*, *Sorangium cellulosum*, and *Streptosporangium roseum* were incorrectly identified by HOPS as present in at least one simulated metagenomic sample, while they did not pass aMeta filtering and were correctly excluded from the resulting output for these samples.

Additionally, we evaluated the performance of aMeta vs. HOPS on a read level (Additional file 1: Fig. S11), by comparing the ground truth and reconstructed read counts for each microbe in each sample. In contrast to HOPS that has a high dropout rate (high false-positive and false-negative counts) and therefore is sensitive to read number threshold (lowering the ~100–300 reads threshold results in a higher number of false-positives), aMeta is more robust in a wide range of read number thresholds due to the additional breadth of coverage filter that reduces the dropout effect. In other words, lowering the ~100–300 reads detection threshold down to ~10–50 reads slightly improves the agreement of aMeta with the ground truth without bringing too many false-positive hits; see Additional file 1: Fig. S11. However, it is important to keep in mind that a microbial hit with only ~10–50 reads (and even higher, up to ~100 reads) would be problematic to authenticate since current gold standard authentication tools such as mapDamage [27] and MaltExtract [23] can reliably operate only on a substantially higher number of reads (at least 200 reads to our experience). Therefore, while specifying more permissive detection thresholds in aMeta might be beneficial for reducing the false-negative rate, it is not recommended due to potential authentication problems. However, if the intent of the user is first to detect a potential organism, before aiming at sequencing more data, then the reduced dropout effect in aMeta compared to HOPS can be taken advantage of. Overall, we conclude that aMeta has a lower detection error compared to HOPS; see Additional file 1: Fig. S5-S11 and Additional file 2: S4 for more details.

Further, we addressed the authentication error of aMeta and HOPS, that is, when a tool, e.g., wrongly reports a microbe as ancient that was actually not simulated to be ancient. For this purpose, we used the authentication scoring systems implemented in aMeta and HOPS. The scoring systems of both tools not only provide a useful ranking of microbial organisms, but can also be used for computing sensitivity and specificity of microbial validation and authentication for benchmarking purposes. We ran aMeta and HOPS with default settings on the simulated ground truth dataset and obtained lists of microbial organisms ranked by the scoring system of aMeta and HOPS, where likely present and ancient microbes received higher scores. Upon visual examination of the native heatmap output generated by HOPS, it became evident that its authentication performance was not optimal (Additional file 1: Fig. S12). More specifically, a few bacteria such as *Rhodopseudomonas palustris*, *Rhodococcus hoagii*, *Lactococcus lactis*, *Brevibacterium*

*aurantiacum*, and *Burkholderia mallei* were mistakenly reported by HOPS to be ancient (as they got the highest scores) in several samples, while they were supposed to be modern according to the simulation's design. The native scoring system of HOPS is based on 3 metrics only (edit distance of all and damaged reads + deamination profile). For more quantitative comparison, it was carefully generalized to match the scoring system of aMeta (see Additional file 2: S5).

Further, we used the scoring systems of aMeta and HOPS to compute receiver operating characteristic (ROC) curves, reflecting the sensitivity vs. specificity of microbial validation and authentication of both tools. The comparison of ROC curves between aMeta and HOPS computed on the simulated ancient metagenomic dataset is presented in Fig. 7. One can observe that for the simulated ground truth dataset, aMeta demonstrates overall higher sensitivity vs. specificity of ancient microbial identification compared to HOPS. This is mainly due to the contribution from the additional evenness of coverage metric (Fig. 4), and better-tuned deamination profile score, which both help aMeta to establish a more informed decision about the microbial presence and ancient status. For example, the species *Burkholderia mallei*, *Brevibacterium aurantiacum*, and *Lactococcus lactis*, which were simulated as modern, obtained high authentication scores from HOPS in some samples, implying they were predicted to be present and ancient. They were, however, correctly ranked low as potential modern contaminants by aMeta. In contrast, the simulated ancient *Salmonella enterica* genome was ranked low by HOPS due to read misalignment (Additional file 1: Fig. S13), while it obtained high scores from aMeta correctly indicating its presence and ancient status; see Additional file 1: Fig. S14. Overall, we conclude that aMeta has a lower authentication error compared to HOPS; see Additional file 2: S5 for more details.

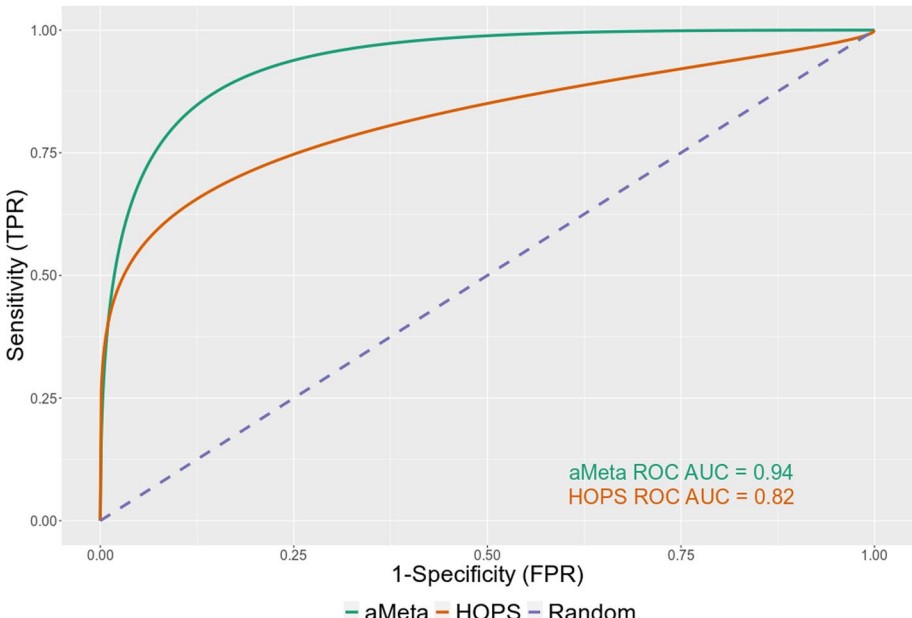

**Fig. 7** Receiver operating characteristic (ROC) curve comparison of authentication scores computed by aMeta and HOPS on simulated dataset

### Effect of database size

An advantage of the full NCBI NT compared to other nucleotide databases is that it provides perhaps the most diverse representation of organisms (both prokaryotic and eukaryotic) across the tree of life. However, due to its large size, it can be memory-demanding for any metagenomic workflow to use this reference database. Therefore, we aimed to investigate to what extent one can reduce the size of the full NCBI NT without compromising the accuracy of organism detection in a metagenomic sample. For this purpose, we first built a few KrakenUniq reference databases (with *k*-mer length 31) varying in size and then used the simulated ancient metagenomic dataset with known microbial composition in order to assess how well KrakenUniq can reconstruct the ground truth depending on the database size.

We found a strong effect of KrakenUniq database size on robustness of microbial detection (Fig. 8). Specifically, after the simulated ancient metagenomic samples have been profiled with KrakenUniq, we filtered the results by depth and breadth of coverage using default thresholds in aMeta: 200 assigned reads and 1000 unique *k*-mers. Next, we computed the Jaccard similarity (intersection over union) between the species detected by KrakenUniq in each database and the ground truth species. We used in total four databases varying in size and content. The smallest database used was the NCBI RefSeq complete microbial genomes, which included 43,767 reference sequences (9155 viral, 440 archaeal, and 34,172 bacterial sequences) comprising nearly 70 billion nucleotide characters. The database can be accessed via https://doi.org/10.17044/scilifelab. 21299541. The largest database was the full NCBI NT, which included 60,179,710 reference sequences containing approximately 230 billion nucleotide characters. This database is available at https://doi.org/10.17044/scilifelab.20205504. The intermediate-size

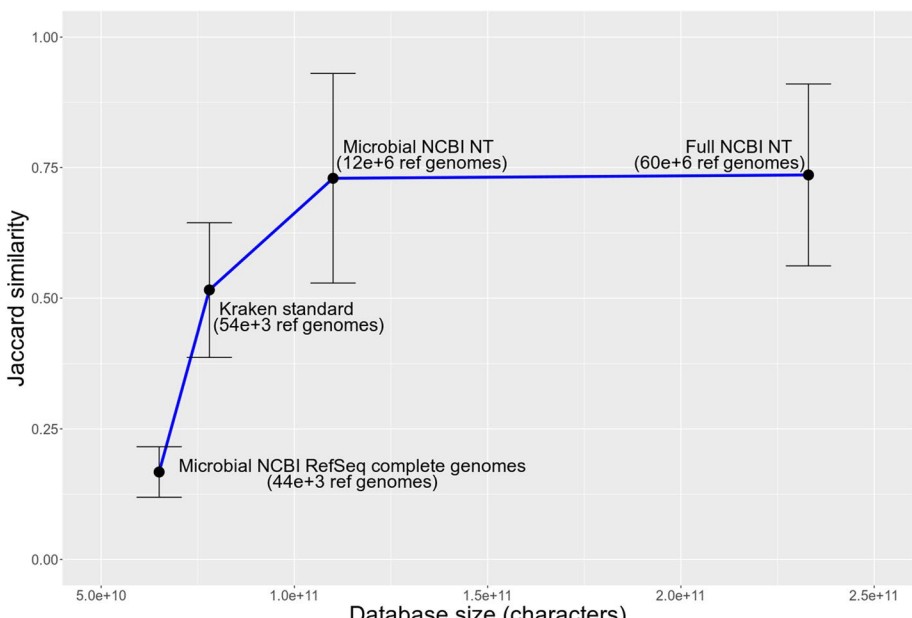

**Fig. 8** Effect of database size on microbial identification with KrakenUniq: Jaccard similarity (intersection over union) metric was computed with respect to a simulated ground truth. Larger databases tend to provide better overlap with the ground truth

databases included the Standard Kraken database (default for both Kraken1 [16] and Kraken2 [17]), and the microbial subset of the full NCBI NT, i.e., Microbial NCBI NT. The Standard Kraken databases included 53,693 reference sequences (11,956 viral, 553 archaeal, and 41,184 bacterial sequences) that together contained nearly 80 billion characters, while the Microbial NCBI NT database included 11,840,243 reference sequences (2,465,945 viral, 17,519 archaeal, 1,737,968 bacterial, 4,530,716 fungal, 1,689,877 protozoa, and 1,398,218 parasitic worms sequences) comprising 110 billion characters.

The smallest NCBI RefSeq complete genomes database gave the lowest Jaccard similarity to the simulated ground truth, just below 0.2, which suggests that this database suffers from low sensitivity of microbial detection which may potentially bias taxonomic profiling of metagenomic samples. We found that increasing the database size resulted in gradual growth of Jaccard similarity, i.e., a better overlap of detected microbial species with the ground truth species. Starting with the Microbial NCBI NT database comprising approximately 110 billion characters, the Jaccard similarity reached a plateau at around 0.75. Although the complete NCBI NT, that included both prokaryotic and eukaryotic reference genomes, was able to further increase the Jaccard similarity metric, the effect was rather marginal (Fig. 8). This database, however, demanded substantially greater RAM resources. Therefore, we concluded that the Microbial NCBI NT provides sufficient accuracy when performing microbial profiling, i.e., including eukaryotic organisms into the database (as it is the case for the full NCBI NT) does not significantly affect the accuracy of microbial detection. Despite the large variation of Jaccard similarity in Fig. 8 indicated by large error bars, which were computed by averaging across samples, the increasing profile of Jaccard similarity as a function of database size is quite clear. Therefore, based on our simulation work, we concluded that larger databases provide higher accuracy of microbial detection, while smaller databases suffer from low sensitivity and may introduce biases into microbial identification in metagenomic samples.

Further, to demonstrate how spurious misalignments may arise when working with small reference databases, we used a random metagenomic stool sample *G69146* from a modern infant from the DIABIMMUNE metagenomic database, Three Country Cohort [37], and aligned it to the *Yersinia pestis* (*Y. pestis*) CO92 reference genome alone. We discovered that nearly 22,000 reads mapped uniquely, i.e., with mapping quality MAPQ > 0 (Fig. 9). Since the sample was from a modern infant who unlikely suffered from the plague, the mapped reads cannot be used as evidence of the presence of *Y. pestis* in the infant's stool sample. Further, visually inspecting the alignments in Integrative Genomics Viewer (IGV) [30], we confirmed that the reads aligned unevenly and demonstrated a high number of multi-allelic single nucleotide polymorphisms (SNPs) implying *Y. pestis* was not a right reference genome for the reads; see Additional file 1: Fig. S2. Assuming that a large fraction of the aligned reads might be of human rather than bacterial origin, and thus misaligned to the *Y. pestis* reference genome due to the absence of a human reference genome in the reference database, we concatenated the hg38 human reference genome with the *Y. pestis* reference genome and proceeded with competitive mapping. We found, however, that adding the human reference genome to the database did not change the number of reads mapped uniquely to the *Y. pestis* reference genome. Next, we assumed that the ~ 22,000 misaligned reads originated from microbial organisms, other than *Y. pestis*, that were phylogenetically closer to *Y. pestis* than to humans.

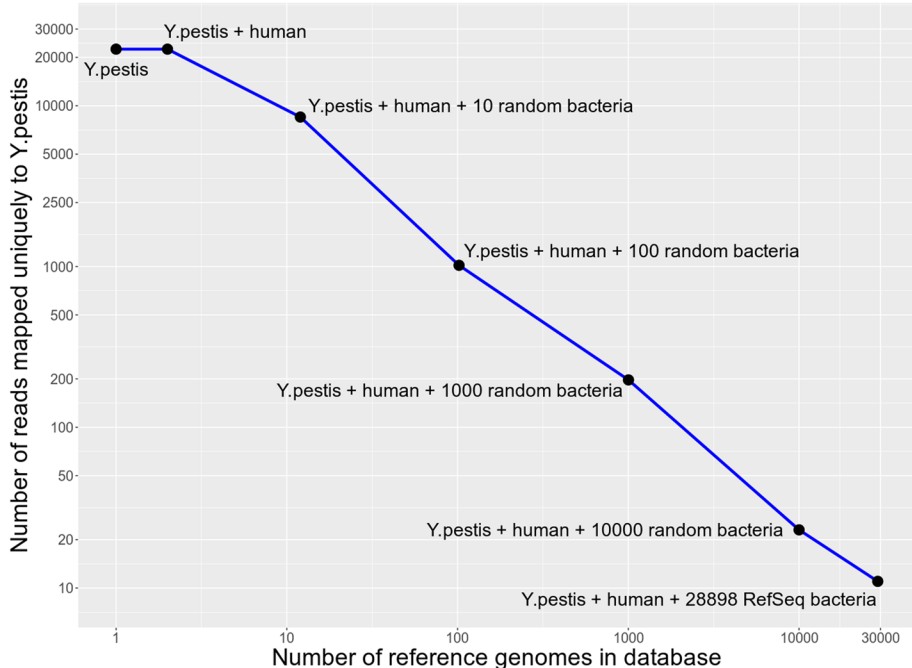

**Fig. 9** Effect of database size on the number of modern metagenomic reads uniquely mapped to *Yersinia pestis* CO92 reference genome. Starting with *Y. pestis* alone, ~ 22,000 reads map uniquely. This number gradually decreases down to only a few reads with the growth of the database, i.e., when the human hg38 reference genome is added, followed by adding 10, 100, 1000, and 10,000 random bacteria, and finally, all available 28898 bacteria from the NCBI RefSeq database. The axes of the plot are log10-scaled

We therefore used sequentially (a) 10 random bacterial reference genomes from the NCBI RefSeq database, (b) 100 random bacterial reference genomes, (c) 1000 random bacterial reference genomes, (d) 10,000 random bacterial reference genomes, and finally (e) all 28,898 bacterial genomes available from NCBI RefSeq for September 2022; concatenated them with *Y. pestis* + hg38; and performed alignments with Bowtie2 to this concatenated reference. We observed a gradual decrease in the number of reads aligned uniquely to *Y. pestis*: from ~ 8500 reads at 10 random bacteria down to only 11 reads at 28,898 bacteria (Fig. 9). This was a substantial decrease compared to the initial ~ 22,000 reads; nevertheless, we still had a few misaligned reads while our expectation was to observe near-zero reads aligning uniquely to the *Y. pestis* reference genome. We believe that the ~ 10 aligned reads can be treated as a noise level and therefore should not be considered as evidence of microbial presence in a metagenomic sample. Thus, the increase in database size, i.e., the number of reference genomes in the database, allowed us to correctly confirm that *Y. pestis* was not present in the modern infant stool sample.

Moreover, we replicated the decreasing profile for the number of (misaligned) reads mapped uniquely to *Y. pestis* by random sampling of bacterial genomes from the Microbial NCBI NT database, this time without the hg38 human reference genome, and for a greater number, i.e., up to 117,000 random reference genomes (Additional file 1: Fig. S15). When sampling reference genomes from the Microbial NCBI NT database, we observed not only a very similar qualitative behavior as in Fig. 9 for genomes from NCBI RefSeq, i.e., bigger databases result in lower numbers of misaligned reads, but also

quantitatively very similar, however slightly higher, numbers as in Fig. 9. We assume that the slightly higher counts of misaligned reads observed for Microbial NCBI NT compared to NCBI RefSeq are related to the difference in the quality of reference genomes in the two databases, i.e., the same number but better quality reference genomes from NCBI RefSeq can "attract" more non-Yersinia reads, and thus result in fewer reads misaligned to *Y. pestis* reference genome.

Overall, we conclude that the database size plays a major role in the robustness of microbial identification in metagenomic analysis. A sufficiently small database, while computationally easier to process, may jeopardize the accuracy of metagenomic analysis and lead to high false-positive and false-negative rates for the detection of microbial species.

### Replication on pathogen-enriched simulated and real datasets

Given the significant interest for pathogen detection in the field of ancient metagenomics, we aimed to replicate the comparison of aMeta and HOPS on another simulated dataset enriched with pathogens, as well as on real ancient metagenomic samples where the presence of microbial pathogens had been previously reported [38].

First, we simulated ten new samples with 5 pathogenic bacteria that were not previously included in the in silico dataset, i.e., *Brucella melitensis*, *Mycobacterium leprae*, *Mycobacterium tuberculosis*, *Treponema pallidum*, and *Vibrio cholerae*. In addition, 3 pathogenic viruses, *Hepatitis B virus*, *Human parvovirus B19*, and *Variola virus*, and one eukaryotic pathogen, *Plasmodium vivax*, were added. All 9 pathogens had been successfully found in aDNA studies [38] and were therefore simulated as ancient, according to the same procedure as described in the "Methods" section (see also Additional file 2: S7). To mimic a typical contamination scenario, we also added 4 modern microbial species from the *Burkholderi* and *Pseudomonas* genera. Moreover, we specifically addressed the limitation of very low pathogen abundance by significantly restricting the number of microbial reads to just 100,000 to 300,000 reads (for comparison, in the main analysis, this number varied from 300,000 to 700,000 reads); see Additional file 2: S7. The simulated dataset enriched for microbial pathogenic species is available at https://doi.org/10.17044/scilifelab.24211584. Additional file 1: Fig. S18 displays the ground truth design of the replication pathogen-enriched dataset, which was subsequently processed using aMeta and HOPS with their default settings. The detection and authentication outcomes from aMeta and HOPS are presented in Additional file 1: Fig. S19 and S20, respectively. While aMeta was able to correctly detect and authenticate all 9 pathogenic species in most of the samples where they were simulated as present (Additional file 1: Fig. S19), HOPS completely missed *Plasmodium vivax* and failed to authenticate *Hepatitis B virus* and *Treponema pallidum* in all simulated samples (Additional file 1: Fig. S20). An example of an authentication plot produced by aMeta for *Plasmodium vivax* correctly detected and authenticated in sample 6 is demonstrated in Additional file 1: Fig. S21. Further, while the authentication scores computed by aMeta and HOPS were broadly comparable, HOPS as in the main analysis tended to designate samples as ancient more readily. For example, a modern contaminant *Burkholderia mallei* erroneously obtained the highest authentication scores from HOPS while it was correctly ranked low by the aMeta

scoring system. Interestingly, we discovered that the *Variola virus* was highly ranked by HOPS in a few samples despite its rather inconclusive deamination profile; see Additional file 1: Fig. S22. In contrast, aMeta assigned relatively low authentication scores to the *Variola virus* in all samples due to the lack of clear enrichment of C/T polymorphisms at the end of the metagenomic reads. We assume that the consistently unconvincing deamination profile for the ancient-simulated *Variola virus* pathogen is related to the poor quality of its reference genome. Finally, to check how sensitive aMeta and HOPS can be when working on a very low coverage data, we compared the numbers of detected reads per simulated pathogen per sample as a function of sequencing depth (Additional file 1: Fig. S23). Both workflows tended to have lower missingness at higher sequencing depth. In other words, aMeta missed 4 pathogens at the total number of microbial reads of 100,000 and 200,000, and 1 pathogen at 300,000, while HOPS missed 4 pathogens at 100,000, 6 at 200,000, and 3 at 300,000 microbial reads. Nevertheless, aMeta demonstrated lower pathogen missingness compared to HOPS, i.e., missed 9 pathogens vs. 13 by HOPS. Overall, we can conclude that aMeta provides satisfactory performance in terms of pathogen detection and authentication even when dealing with low pathogen abundance.

In addition to the simulated pathogen-enriched dataset, we tested the performance of aMeta on real microbial pathogens from 4 ancient metagenomic studies: *Helicobacter pylori* [39], *Borrelia recurrentis* [40], *Brucella melitensis* [41], and *Tannerella forsythia* [42]. In total, metagenomic sequencing data from 36 libraries were tested. Although no ground truth information was available for the tested libraries, we assumed pathogen presence in all of them. aMeta was successful in detecting the reported pathogens in all but 4 tested libraries (Additional file 3: Table S1 and Additional file 1: Fig. S24), which did not pass the default thresholds of aMeta: 200 assigned reads and 1000 unique *k*-mers. Nevertheless, we discovered that two of them, i.e., the libraries ERR1094778 and ERR1094779 [39], had extremely low and uncommon sequencing depth, i.e., approximately 500,000 reads, and that the libraries ERR1094791 and ERR1094792 from the same study corresponded to muscle tissue, where, in fact, no evidence of *Helicobacter pylori* had been found in the original study [39]. Further, we found that *Borrelia recurrentis* was discovered by aMeta in all libraries from the study [40]; however, very few reads were assigned to the *species* rank implemented by default in aMeta. In fact, the vast majority of reads were assigned to *Borrelia recurrentis A1*, complete genome, GeneBank accession number CP000993.1, which had a *sequence* rank in the NCBI taxonomy used in this study. This suggests that rank filters of aMeta can be improved by adding *sequence* and possibly *no rank* categories. On the other hand, this also shows the importance of large and diverse databases used by aMeta, where inclusion of all available microbial strains can be critical for the detection of a rare pathogen. Importantly, as it is demonstrated in Additional file 1: Fig. S24, the default thresholds for depth and breadth of coverage of aMeta provide fair sensitivity of pathogen discovery in a wide range of library sizes. More specifically, aMeta is capable of recovering pathogens even in libraries sequenced at depth as low as 8 mln reads and potentially even lower. Overall, we conclude that aMeta successfully confirmed the presence and absence of corresponding pathogens in 34 out of 36 analyzed libraries from 4 different shotgun metagenomic studies [39–42], which implies a satisfactory accuracy of performance on real data.

## Discussion

While the methodology of traditional ancient genomics reached maturity some time ago, there still does not seem to be a profusion of analytical tools to perform ancient microbiome analysis, presumably because the latter is a much younger field. Currently available ancient metagenomics workflows such as MALT [20], HOPS [23], and nf-core/eager [33]—the latter internally using HOPS—are sensitive to the choice of reference database and are therefore not always optimal in terms of sensitivity vs. specificity balance of microbial detection. Furthermore, when performing reference-based microbiome profiling, the size of a reference database becomes an important factor as large databases should provide more unbiased identification of present microbes. If the reference database is not large enough, there is a risk, first, of not identifying a microorganism that is not present in the database (Fig. 8), and second, of erroneously identifying a microorganism in the database that happens to be phylogenetically close to another microbe truly present in the sample but not included in the database; see the example in Fig. 9. However, current analytical tools such as MALT [20], HOPS [23], and nf-core/eager [33] can only be run on reference databases of limited size; see Additional file 2: S2 for more details. There is, therefore, a current need for alternative, more accurate, and memory-efficient ancient metagenomics profiling workflows that can query metagenomic samples against large reference databases.

In this study, we proposed a novel *a*ncient *Meta*genomic workflow, aMeta, which has a number of advantages over other analytical frameworks in the field. The workflow is based on recent advances in the field of metagenomics and provides a list of ancient microbes robustly detected and authenticated based on multiple quality metrics with minimal interference from the user. Unlike other typical workflows that often merely combine heterogeneous bioinformatic tools, aMeta was designed to answer a specific research question, which is the robust identification of ancient microbial organisms with optimal sensitivity and specificity of detection and authentication. Therefore, while at first glance our workflow can be seen as a combination of *k*-mer-based classification of microbial DNA fragments via KrakenUniq and LCA-based alignment via MALT, it implements in fact a number of additional features that (1) harmonize the outputs of KrakenUniq and MALT and make them work coherently, (2) minimize the amount of manual post-processing work, (3) optimize memory usage, and (4) ensure users obtain an easy to grasp and highly accurate overview of the microbial composition of the query samples.

More specifically, aMeta uses taxonomic pre-screening with KrakenUniq against a large reference database to inform LCA-based alignment analysis with MALT. Initial unbiased pre-screening against large databases becomes computationally feasible thanks to the recent low-memory development of KrakenUniq [22]; meaning, provided that a reference database has been already built and is of a reasonable size, taxonomic classification can be performed on virtually any computer, even a laptop, irrespective of the database size. This new development opens up exciting opportunities for truly unbiased pre-screening with KrakenUniq, followed by alignment, validation, and authentication with MALT, as implemented in our workflow. This approach reduces the memory requirements for MALT during the follow-up step, as MALT's memory usage can be minimized by selecting likely present microbial organisms detected by KrakenUniq in

the initial pre-screening step. This substantially reduces the memory consumption of MALT.

Effectively, our computer memory benchmarking shows that aMeta consumed barely half the RAM compared to HOPS when processing 10 simulated ancient metagenomic samples (Fig. 5). The memory gain can be explained by two factors. First, despite a larger database used by aMeta (microbial version of NCBI NT + human + complete eukaryotic genomes, the reference sequences occupy ~ 300 GB of disk space) than by HOPS (complete microbial genomes from NCBI RefSeq database, the reference sequences occupy ~ 60 GB of disk space), the recent fast and low-memory development of KrakenUniq [22] was able to handle the larger database more efficiently and to use less memory compared to MALT, which is the implicit engine of HOPS. Second, as a result of pre-screening with KrakenUniq, the dynamically built MALT database had a reduced size compared to the MALT database used for HOPS. In other words, the MALT step in aMeta is not a screening *per se* but a follow-up after a KrakenUniq pre-screening. Thus, it can be performed using a reduced database, unlike HOPS, which is a screening pipeline by design, where in order to obtain an unbiased microbial detection, one has to use a large MALT database which imposes a hard computational resource burden as indicated in Fig. 5. More specifically, we were able to run HOPS with our smallest database (NCBI RefSeq complete microbial genomes) only on computer nodes with at least 1 TB of RAM. In contrast, aMeta was capable of running on 512 GB, and even 256 GB nodes, despite using a much larger (Microbial NCBI NT) reference database. Thus, aMeta demonstrates a substantial reduction in memory load.

Importantly, the memory gain of our workflow does not compromise the accuracy of microbial detection and authentication. Instead, as shown in Figs. 6 and 7, aMeta has a better sensitivity vs. specificity balance for both microbial detection and authentication compared to HOPS in a wide range of target reads thresholds. On the one hand, the superior sensitivity of aMeta comes from a larger reference database used by KrakenUniq compared to the one used by HOPS. In essence, including more microbial organisms into the reference database enables their discovery in query samples. On the other hand, the superior specificity of aMeta is primarily due to robust filtering based on the evenness of coverage applied to candidate microbes. In other words, aMeta does not only rely on the number of reads mapped to a reference genome of a microbial candidate, as does essentially HOPS, but considers the spread of aligned reads across the reference genome as an ultimate criterion of microbial presence. While the evenness of coverage is a crucial metric, aMeta also generates a few other quality metrics such as deamination pattern, edit distance, PMD scores, read length distribution, average nucleotide identity (ANI), and depth of coverage (see Fig. 4) and combines them into a score that can be used to rank microbial candidates to get a robust overview of the ancient microbiome. A graphical overview of the scores per sample and per microbial candidate (Additional file 1: Fig. S4) allows users to quickly understand the ancient microbial composition of the query samples and make informed decisions about further sequencing or targeted enrichment strategies.

Furthermore, unlike HOPS, aMeta was not designed to serve solely for pathogen screening, but can very well operate as a general ancient microbiome profiling framework, i.e., covering a much broader spectrum of microbial organisms. However,

screening for pathogen DNA in archaeological remains can be one possible application of aMeta, which utilizes for this purpose a comprehensive list of microbial pathogens that was custom built based on literature. Since screening for microbial pathogens is typically performed on a very limited number of target reads, the outcome of aMeta might be sensitive to the filtering thresholds applied. The major filter of aMeta is the breadth of coverage, which is approximated via the number of unique *k*-mers (Additional file 1: Fig. S1). By default, aMeta requires at least 1000 unique *k*-mers per taxon for detection. The number of this order of magnitude was recommended in the original KrakenUniq publication [21] and corresponds to approximately $\sim 50$ non-overlapping reads mapping to the taxon reference. Indeed, since our KrakenUniq databases were built with a length of *k*-mer equal to $k = 31$, and provided that a typical length of a fragmented aDNA would be $L \sim 50$ bp, this would result in $L - k + 1 = 50 - 31 + 1 = 20$ k-mers. If all the *k*-mers are unique, i.e., they correspond to non-overlapping reads, there should be at most 50 non-overlapping reads, which would provide $50 \times 20 = 1000$ unique *k*-mers. Therefore, in a broad sense, the breadth of coverage of 1000 unique *k*-mers indirectly requires a depth of coverage of at least 50 assigned reads for a typical aDNA fragmentation. However, it is important to keep in mind that although 50–100 assigned (non-overlapping) reads would be enough for detection, this small number of reads would be problematic to authenticate with, e.g., mapDamage [27]. Taking into account this limitation, we chose to use 200 assigned reads specific to a taxon as a default threshold for depth of coverage in aMeta. The choice of 200 assigned reads and 1000 unique *k*-mers as defaults was justified from different angles by (1) simulation benchmarking (Fig. 6, Additional file 1: Fig. S10 and Additional file 1: Fig. S11) and (2) 36 real ancient shotgun metagenomic libraries from 4 different studies reporting pathogens (Additional file 1: Fig. S24, and 3) exhaustive empirical testing in a large dataset of over 1200 ancient metagenomic libraries [35, 36]. Nevertheless, it is important to emphasize that the depth and breadth of coverage thresholds are optional in aMeta and can be adjusted depending on the project goals and data quality. For our ancient metagenomic profiling projects, we typically use $\sim 100 - 300$ assigned reads and $\sim 500 - 1500$ unique *k*-mers, and we encourage the users of aMeta to experiment with the depth and breadth of coverage filters, and tune them for their particular projects.

Despite the memory efficiency of aMeta for running ancient metagenomic analysis, it assumes that the KrakenUniq database and Bowtie2 index have been built prior to the analysis, which is a computer memory-demanding process. For example, building a full NCBI NT, KrakenUniq database (with *k*-mer length 31) in December 2020, required up to 4 TB of RAM. Therefore, with the aMeta release, we make a few large pre-built KrakenUniq databases and Bowtie2 indexes publicly available for the community; see the "Availability of data and materials" section. Currently, aMeta users do not have to build their own databases and indexes, which is a time- and memory-consuming process, but can freely download the large pre-built databases and use them for their analysis.

Finally, it is important to mention that our workflow follows the standards of reproducible data analysis via the workflow management system, Snakemake [24]. The Snakemake implementation of aMeta not only facilitates reproducibility and scalability of the data analysis, but also allows for seamless integration in high-performance computer

(HPC) cluster and cloud environments; see the "Methods" section and Additional file 2: S6.

### Limitations and planned extensions of aMeta

Despite offering advantages in terms of accuracy and resource usage, aMeta has a few limitations which are worth mentioning.

First, aMeta uses a reference-based approach for the discovery of microbial organisms in metagenomic samples. This implies that only organisms included in a reference database can be found in a sample. Therefore, a current disadvantage of aMeta is that it is not able to discover unknown microbial organisms for which there is no reference genome generated yet. This problem, however, is not specific to aMeta but rather to the approach as it is also valid for other tools following reference-based strategy such as MALT [20], HOPS [23], and nf-core/eager [33].

An alternative approach widely used in modern metagenomics [43–46], and reaching maturity in ancient metagenomics [47], is the de novo assembly of microbial contigs. With this method, no prior information about potential microbial candidates is required, and reference genomes can be reconstructed virtually for any microbe present in a sample. This process however typically requires high coverage, i.e., deep sequenced samples, which might be problematic for palaeogenetics where usually a very limited amount of ancient DNA can be extracted from archaeological artifacts. Another difficulty comes from the ancient DNA damage [28] that, in addition to sequencing errors, complicates the de novo assembly process and can lead to the formation of chimeric contigs [48] which could greatly influence the downstream analysis.

A de novo assembly module (not presented in this article) written in Snakemake is currently being tested in our lab, and we plan to add it to the workflow in a future release of aMeta. This way, aMeta will leverage the power of classification, alignment, and de novo assembly which can complement each other and provide a more informative overview of microbial composition in ancient metagenomics samples.

Another planned extension of the aMeta workflow is a special mode for working with ancient environmental and sedimentary DNA, an area of palaeogenetics that has experienced a rapid growth [49]. One challenge here to overcome is the fine-tuning of the aMeta workflow to deal with large eukaryotic reference genomes such as plant and animal genomes. For this purpose, using the non-redundant NCBI NT database may not be optimal as it contains eukaryotic reference genomes, which are typically of poor quality and far from complete. Our preliminary testing shows that the large variation in quality of reference genomes across eukaryotic organisms in the NCBI NT database can lead to severe biases in taxonomic assignment of metagenomic reads, where spurious taxa can be detected merely because they have better quality (more complete) reference genomes compared to homologous taxa that are in fact present in the sample.

Further, although the internal default filters used by aMeta are well-tuned and seem to demonstrate good performance for a vast majority of aDNA samples [35, 36], we are working on developing a strategy for self-adjusting the filters depending on the nature and quality of aDNA samples. For example, viruses have typically small reference genomes, and hence, very few aDNA reads aligned to them. Therefore, hard

filtering thresholds that are currently implemented in aMeta might miss rare members of the microbial community and need further tuning.

Next, although the pre-screening step with KrakenUniq implemented in aMeta substantially reduces the amount of memory needed for performing MALT alignments, we found that large input *fastq*-files (> 500 million sequenced reads) from deeply sequenced samples, or alternatively, a large number (> 1000) of medium-size input *fastq*-files can still result in a severe memory burden for the MALT step that might consume over 1 TB of RAM, even though KrakenUniq is rather insensitive toward the input file size. Therefore, we do not currently recommend merging *fastq*-files from different sequencing libraries corresponding to the same sample as it is often done in genomics projects, but we advise processing *fastq*-files individually unless one has access to very large computer nodes.

Finally, aMeta may currently not be as fast as HOPS when extensive multi-threading is available. Indeed, in our benchmarking (Fig. 5), HOPS was nearly twice faster than aMeta, i.e., 250 min vs. 500 min, when both were using 20 threads. Note that HOPS was however slower on 1 thread, i.e., 650 min. The advantage of HOPS in speed is not surprising because it uses a pre-built MALT database (based on NCBI RefSeq complete microbial genomes), while the time-consuming dynamic building of a MALT database is a part of the aMeta run. In addition, a few other essential but time-consuming modules of aMeta such as KrakenUniq, Bowtie2, and mapDamage are not part of the HOPS pipeline, which gives an additional speed advantage to HOPS. Nevertheless, we are currently developing several optimization schemes that can potentially improve the speed of aMeta in the future release. We are also communicating with the nf-core developer team and planning to integrate aMeta to nf-core/eager for better versatility, maintenance, and efficiency.

Nevertheless, already in the current state, the aMeta workflow gives clear advantages compared to the state-of-the-art HOPS in terms of accuracy and computer memory usage, which can potentially improve the quality of computational analysis in the ancient metagenomics field and which we hope will be appreciated by the ancient DNA research community.

## Conclusions

aMeta is a novel computational workflow for ancient metagenomics that aims at improving analysis accuracy and optimization of computational resources. aMeta combines the advantages of a *k*-mer-based taxonomic classification approach in sensitivity and of a Lowest Common Ancestor (LCA) alignment approach in specificity of microbial discovery and authentication. On our simulation benchmark, aMeta demonstrated better performance in terms of accuracy and memory load compared to HOPS which is currently a gold standard approach in the area of ancient metagenomics. We also evaluated aMeta on data from a few ancient shotgun metagenomics studies, as well as previously in multiple aDNA projects in our lab, where aMeta consistently demonstrated accurate and computationally feasible performance. Therefore, aMeta is likely to be of broad utility for the ancient metagenomics field and aDNA research community.

## Methods

### Snakemake implementation of aMeta

aMeta was written using the Snakemake language for workflow management [24] which ensures reproducibility of the ancient metagenomic data analysis performed by aMeta. The workflow together with installation instructions, examples of command lines, documentation, vignettes, and test dataset can be accessed from GitHub https://github.com/NBISweden/aMeta [50] and Zenodo https://zenodo.org/record/8354933 [51]. aMeta was developed as a collection of Snakemake rules listed in a *Snakefile* according to the Snakemake best practices (https://snakemake.readthedocs.io/en/stable/snakefiles/best_practices.html#snakefiles-best-practices) and workflow template (https://github.com/snakemake-workflows/snakemake-workflow-template). Snakemake automatically determines the execution order of the rules following a Directed Acyclic Graph (DAG) of jobs that can be automatically parallelized (Additional file 1: Fig. S17). The workflow and each separate rule can be installed via *conda*, a package manager (https://conda.pydata.org/). Main configuration options of aMeta, for example KrakenUniq filtering thresholds, can be specified for a particular dataset in the *config.yaml* file located within the *config* directory. The Snakemake configuration of aMeta can be easily adapted to both local computers and high-performance computers (HPC).

### *Quantifying authentication information* via *aMeta's authentication scores*

The scoring system of aMeta was developed for user convenience as a fast visual overview of ancient microbial species present in each metagenomic sample. Since it may be time-consuming, and sometimes not even feasible to visually inspect all the quality metrics presented in Fig. 4 for each sample and each detected microbe, aMeta implements a special scoring system to quantify the authentication and validation metrics. The scoring system of aMeta represents a sum of eight validation and authentication metrics computed on the LCA alignments delivered by MALT: (1) deamination profile, (2) evenness of coverage, (3) edit distance for all reads, (4) edit distance for damaged reads, (5) read length distribution, (6) PMD scores distribution, (7) number of assigned reads (providing information on the depth of coverage), and (8) average nucleotide identity (ANI). Each metric can add$+1$ to the total sum except for the evenness of coverage that can add$+2$, as aMeta considers it to be the most crucial to validate microbial presence, and for the deamination profile, which can add up to$+2$ (both $5'$ and $3'$ ends count independently), as we assume it to be the ultimate criterion of ancient status of a microbe. Therefore, the range of authentication scores a microbe can obtain varies from a minimum of 0 to a maximum value of 10; see Additional file 2: S5 for more details.

### Simulation of ancient metagenomic data

We used the gargammel tool [34] to simulate 10 metagenomic samples with varying human and microbial composition. Both host-associated and contaminant reads were present in the simulated samples. In total, 35 microbial species (31 bacteria, 2 amoeba, 1 fungus, and 1 algae) commonly found in our ancient metagenomic projects [35, 36] were simulated with varying abundance across the samples. The abundance of each microbe

in a metagenomic sample was set randomly, and the addition of host-associated and contaminant fractions sums up to 1 per sample. We simulated reads belonging to 18 ancient and 17 modern microbes. The list of simulated microbial organisms is shown below:

> *Ancient: Campylobacter rectus, Clostridium botulinum, Enterococcus faecalis, Fusarium fujikuroi, Mycobacterium avium, Mycolicibacterium aurum, Neisseria meningitidis, Nocardia brasiliensis, Parvimonas micra, Prosthecobacter vanneervenii, Ralstonia solanacearum, Rothia dentocariosa, Salmonella enterica, Sorangium cellulosum, Streptococcus pyogenes, Streptosporangium roseum, Yersinia pestis, Bradyrhizobium erythrophlei*
>
> *Modern: Acanthamoeba castellanii, Aspergillus flavus, Brevibacterium aurantiacum, Burkholderia mallei, Lactococcus lactis, Methylobacterium bullatum, Micromonas commoda, Micromonospora echinospora, Nonomuraea gerenzanensis, Pseudomonas caeni, Pseudomonas psychrophila, Pseudomonas thivervalensis, Vermamoeba vermiformis, Rhodococcus hoagii, Rhodopseudomonas palustris, Mycobacterium riyadhense, Planobispora rosea*

For ancient microbial reads, we implemented deamination/damage pattern with the following Briggs parameters [27, 28] in gargammel: *-damage 0.03,0.4,0.01,0.3*. The simulated ancient reads were fragmented and followed a log-normal distribution with the following parameters *–loc 3.7424069808 –scale 0.2795148843*, which were empirically determined from the *Y. pestis* reads in another project [35]. Illumina sequencing errors were added with the ART module of gargammel to both modern and ancient reads. Finally, Illumina universal sequencing adapters were used, which resulted in 125 bp long paired-end reads. Each simulated metagenomic sample contained 500,000 ancient and 500,000 modern DNA fragments. The total microbial DNA fraction varied as 0.7, 0.7, 0.7, 0.5, 0.5, 0.5, 0.4, 0.3, 0.3, 0.3 between samples 1 and 10, i.e., the microbial DNA percentage varied between 30 and 70% per sample, with the remainder of the reads belonging to human DNA. The codes used for generating ground truth microbial abundances as well as simulating ancient metagenomic reads are available on GitHub: https://github.com/NikolayOskolkov/aMeta via Zenodo https://doi.org/10.5281/zenodo.8130819.

**Computation of evenness of coverage**

The evenness of coverage plot in Fig. 4 is computed by aMeta from BAM alignments, produced by MALT or Bowtie2, by splitting the reference genome in 100 bins, counting the number of reference positions covered by at least one aligned read within each bin, and normalizing this count by the total number of genomic positions in each bin. Technically, this procedure is performed via the *samtools depth* command from SAMtools with *-a* flag [29]. This command produces a file reporting the number of reads covering all positions of the reference genome. By definition, the number of genomic positions covered at least once and normalized by the total number of genomic positions represents the breadth of coverage. Therefore, the evenness of coverage plot produced by aMeta in Fig. 4 can be considered as a local breadth of coverage computed in each bin across the reference genome. It is expected that a good evenness of coverage has few or no bins with a value of zero. Therefore, although the genome-wide average breadth

of coverage can be very low in shotgun ancient metagenomic studies due to overall low sequencing depth, an even distribution of the reads provides a good hint of microbial presence in a sample, which can be followed up by deeper sequencing or target enrichment (capture) experiments. See Additional file 2: S3 for more details.

## Supplementary Information

---

**Additional file 1: Supplementary figures Fig. S1-24**, additional figures with technical information not included in the main text.

**Additional file 2: Supplementary information S1-S7**, other technical details not included in the main text and Methods section [52–56].

**Additional file 3: Supplementary Table S1.** summary statistics of pathogen detection by aMeta in 4 real datasets.

**Additional file 4.** Review history.

---

### Acknowledgements

James A. Fellows Yates, Alexander Herbig, Felix Key, Nicolás Rascovan, Maxime Borry, Alexander Hübner, Irina M. Velsko, Alina Hiss, Gunnar Neumann, and Christina Warinner are greatly acknowledged for providing valuable feedback on the design and technical details of the workflow. We thank Åke Sandgren at SNIC for his assistance with cluster implementation aspects, which was made possible through application support provided by SNIC. We also thank Stephan Nylinder and SciLifeLab Data Centre for their help with depositing data and codes in public repositories.

### Review history

The review history is available as Additional file 4.

### Peer review information

### Authors' contributions

NO, PU, CM, and ZP designed and developed the workflow. TN, EK, NB, MV, MK, and EA extensively tested the workflow and suggested improvements. TvdV, LD, and AG discussed the results. NO, NB, and ZP wrote the manuscript. All authors read and approved the final manuscript.

### Authors' Twitter handles

Twitter handles: @ZoePochon (Zoé Pochon), @nora_bergfeldt (Nora Bergfeldt), @emrahkirdok (Emrah Kırdök), @T_vd_Valk (Tom van der Valk), @ezgimou (Ezgi Altınışık), @love_dalen (Love Dalén), @AndersGother (Anders Götherström), @clamirabello (Claudio Mirabello), @unnebe (Per Unneberg), @NikolayOskolkov (Nikolay Oskolkov).

### Funding

 NO, PU, and CM are financially supported by Knut and Alice Wallenberg Foundation as part of the National Bioinformatics Infrastructure Sweden at SciLifeLab. The computations were enabled by resources provided by the Swedish National Infrastructure for Computing (SNIC), partially funded by the Swedish Research Council through grant agreement no. 2018–05973, in particular projects: SNIC 2021/5–335, SNIC 2021/6–260, SNIC 2022/5–100, SNIC 2022/6–46, SNIC 2022/22–507, SNIC 2022/23–275, and Mersin University BAP project 2019–3-AP3-3729. AG is financially supported by the Swedish Research Council (VR; 2019–00849).

### Availability of data and materials

The workflow is publicly available at https://github.com/NBISweden/aMeta[50] and also deposited in a Zenodo repository https://zenodo.org/record/8354933[51]. The non-redundant NCBI NT KrakenUniq database can be accessed at the SciLifeLab Figshare following the address: https://doi.org/10.17044/scilifelab.20205504, and the microbial version of NCBI NT combined with human and complete eukaryotic reference genomes can be accessed via SciLifeLab Figshare at https://doi.org/10.17044/scilifelab.20518251. Next, the smallest KrakenUniq database used in this study, i.e., the KrakenUniq database based on complete genomes of microbial NCBI RefSeq, is available at https://doi.org/10.17044/scilifelab.21299541. Further, the Bowtie2 index of NCBI NT is publicly available through SciLifeLab Figshare at https://doi.org/10.17044/scilifelab.21070063, and the pathogenic microbial subset of this index can be accessed via the SciLifeLab Figshare at https://doi.org/10.17044/scilifelab.21185887. Codes for computer simulations and other scripts used for this article can be accessed at https://github.com/NikolayOskolkov/aMeta and via Zenodo https://doi.org/10.5281/zenodo.8130819. Finally, the simulated metagenomic dataset with known ground truth used for benchmarking aMeta against HOPS is accessible via SciLifeLab Figshare link https://doi.org/10.17044/scilifelab.21261405, and the simulated dataset enriched for microbial pathogenic species is available at https://doi.org/10.17044/scilifelab.24211584.

## Declarations

### Ethics approval and consent to participate

Not applicable.

**Consent for publication**
Not applicable.

**Competing interests**
The authors declare that they have no competing interests.

**Author details**
[1]Centre for Palaeogenetics, Stockholm, Sweden. [2]Department of Archaeology and Classical Studies, Stockholm University, Stockholm, Sweden. [3]Department of Zoology, Stockholm University, Stockholm, Sweden. [4]Department of Bioinformatics and Genetics, Swedish Museum of Natural History, Stockholm, Sweden. [5]Department of Biotechnology, Faculty of Science, Mersin University, Mersin, Turkey. [6]Ancient DNA Unit, Science for Life Laboratory, Stockholm, Sweden. [7]Ancient DNA Unit, Science for Life Laboratory, Uppsala, Sweden. [8]Human-G Laboratory, Department of Anthropology, Hacettepe University, 06800 Beytepe, Ankara, Turkey. [9]Department of Physics, Chemistry and Biology, Science for Life Laboratory, National Bioinformatics Infrastructure Sweden, Linköping University, Linköping, Sweden. [10]Department of Cell and Molecular Biology, Science for Life Laboratory, National Bioinformatics Infrastructure Sweden, Uppsala University, Uppsala, Sweden. [11]Department of Biology, Science for Life Laboratory, National Bioinformatics Infrastructure Sweden, Lund University, Lund, Sweden.

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

## 

