## [**Additional file 4.** Review history. · Genome Biology]

Review History

First round of review

Reviewer 1

Were you able to assess all statistics in the manuscript, including the appropriateness of statistical tests used? Yes: Statistics look good to me.

Were you able to directly test the methods? Yes.

Comments to author:

The authors present the development of a workflow for ancient metagenomics, aMeta. The aim of this pipeline is to overcome the disadvantages of the most used tools for taxonomic screening of aDNA sequences and to allow a more robust identification of ancient microorganisms. To achieve this, the authors combine two different approaches, a k-mer-based taxonomic classification performed with KrakenUniq, and a subsequent alignment-based classification approach with MALT, which is done using a dynamically built database within the same workflow. Seven additional validation metrics are included to reduce detection and authentication errors. The authors show that their workflow has a better computational performance and higher specificity and detection accuracy than the commonly used tools.

aMeta is proposed to be a memory-efficient and highly specific taxonomic classification workflow. This workflow is implemented in Snakemake, which improves its reproducibility and optimizes its performance in HPC environments. A main improvement of aMeta is the creation of project-specific databases based on the presence of certain candidate species. This step, together with additional validation metrics commonly used in taxonomic profiling analyses, improves the specificity of the classification while optimizing the computational resources needed.

I believe that this workflow is a good contribution to the aDNA field which will be well welcomed by the aDNA community. Although there is space for further improvements, some of which are also acknowledged by the authors in the manuscript, their proposed workflow is a valuable tool to be considered.

Points of consideration.

I think that it would be a good idea to add more explicit remarks and examples of the performance of aMeta in function of the number of microbial reads detected. Figure 3 and 5 provide some hints about it, but the real example shown in Figure 4 corresponds to a rather atypical case (Gok2) in which almost the whole genome of the pathogen was covered in the sequencing data. How will this look like, for instance, for the Gok4 individual that contains only ~2000 mapping reads (compared to the ~200K reads in Gok2)?

The authors explored 18 ancient microbial species, but I believe that some additional pathogens that have been already successfully recovered in ancient DNA studies could be also included and tested too. I have then a following question regarding the detection and authentication capacity of aMeta for all these different ancient species when the number of target reads is limiting? (i.e.,

when the number of hits is closer to the 300 reads & 1000 k-mers threshold used by aMeta). *Y. pestis* is easy to detect and authenticate, but what about the other microbial species?

I think this detection and authentication threshold is very important to report more clearly because in the majority of cases the conservation of ancient pathogen genomes is very poor, so screenings must be performed on a very limited number of target reads, and decisions must be made on this limiting information to move into further sequencing or targeted enrichment strategies.

One of the main strengths of aMeta is the use of a dynamically generated database based on a subset of selected species. These species are selected based on seven validation parameters such as breadth of coverage and postmortem damage. However, during the benchmarking process, aMeta reported a slightly higher number of false positive hits, 14, compared to HOPS, 11 (supp fig 11). My main concern regarding this result, is that even after applying the filters proposed by the authors, HOPS, which doesn't use these filters, has a higher sensitivity for this type of errors. How do authors explain this?

The authors mention that the k-mers unique to each taxa could be considered equivalent to the breadth of coverage information. I think it will be nice if they could develop this concept a bit further in the manuscript. For instance, how it conceptually relates to the breadth of coverage that can be calculated by SAMtools from a BAM file? Also, if I understand well, regions that are conserved among taxa are thus not counted because they are not unique to each taxa, right? Do the authors believe that masking such non-unique k-mers could have come confounding effects?

Some additional Minor comments:

Introduction: The use of the term "endogenous" microbial communities. Does the definition of "endogenous" work well with the context in which it is used in the text? I've seen other terms, such as "targeted", or "of interest", or "host-associated", I would just double check that the term endogenous is ok here.

Intro, page 4, line35: I would not say that authentication error is specific of ancient metagenomics. This can be also an issue with modern samples, e.g., when detecting pathogens in clinical samples.

Supplementary information Page 3, line 8; Suppl Fig 2. The authors show that the Jaccard similarity reached a plateau starting with the Microbial NCBI NT database comprising around 110 billion characters. As one normally considers the numbers of genomes at the moment of creating a database instead of the number of characters, I think it would be nice to provide information regarding the number of genomes from each type were included in this database: bacteria, viruses, archaea, fungi, protozoa and parasitic worms.

Supplementary information Page 4, line 11; Suppl Fig 3. When analyzing the impact of the database size, the authors observed that concatenating the hg38 human genome did not have an impact in the number of misaligned reads. As they mentioned in the manuscript, it is expected that with bigger databases the number of misaligned reads would get closer to zero. If possible, it

would be interesting to see if this holds true after removing the hg38 genome and increasing the number of microbial genomes from the Microbial NCBI NT, keeping it in a reasonable amount of genomes to be able to run with the same amount of resources.

Page 12, first line - Typo ".." Instead of "."

Figure 3. Use of "kmers" instead of "k-mers" as in the rest of the manuscript.

Reviewer 2

Were you able to assess all statistics in the manuscript, including the appropriateness of statistical tests used? Yes: All statistics and metrics used (e.g. IoU and F1 scores) are appropriate.

Were you able to directly test the methods? Yes.

Comments to author:

This is an excellent paper that presents a useful workflow for the taxonomic classification of ancient metagenomic shotgun reads. While the paper does not present a new program per se, it does combine existing software (i.e. krakenuniq, MALT) in a unified profiling workflow that minimizes false assignments while reducing memory requirements, which is much needed due to the ever increasing size of ancient metagenomic datasets (both in terms of the number of samples, as well as the amount of sequencing data). We have been using an earlier version of the same workflow in our lab and we agree with the authors that it is much superior to other workflows in the field, both in terms of accuracy and computational requirements, and I am convinced that it (or permutations of it) will be widely adopted by researchers in the field.

I have very little, if anything to object to the manuscript as the methods are well described and the results presently very clearly. But I do have a few suggestions that I think might improve the manuscript (listed in order that they appear in the manuscript):

- 1) The authors opted to go for KrakenUniq over other k-mer based classifiers and while I agree with that choice (as it has several advantages over say Kraken2) it might be worth pointing out what those advantages are.
- 2) Similarly, I would be interested to know why the authors opted to go for MALT over other mappers (e.g. minimap2 etc)?
- 3) A very minor point but on page 6 of the manuscript, the authors discuss why detecting microbial organisms solely based on "depth of coverage (or simply coverage)" might lead to false positive assignments. While I entirely agree with this point, it might be good to explain what exactly is meant by "coverage" (given the amount of confusion that exists in the field around terms like depth and breadth of coverage etc.). Simply saying coverage, might be unhelpful, I think. For clarity's sake, you could also introduce the term "evenness of coverage" in that same paragraph.

4) On page 7, you briefly discuss the effects of DB size on the rates of false assignments. I think this is a very important point to make and I wonder if it might be worth discussing some of the results in more detail rather than banning them to the supplemental information.

5) Again a very minor point, but on page 8 you write that "KrakenUniq ... cannot control the authentication error". I know what you mean but I think it could perhaps be phrased a bit more clearly, so maybe rephrase or perhaps just drop "cannot control the authentication error because"?

6) On page 9, you write: "that would otherwise be technically impossible for MALT to build". I think maybe "handle" might be a better word here?

7) On page 11, you list the seven metrics you use to validate a true ancient microbe but what about others like average nucleotide identity (ANI)?

8) On page 13, you write that the KrakenUniq and HOPS microbial abundance matrices were filtered using different thresholds for the number of assigned reads and later on in the same paragraph you write that "the default filtering thresholds were empirically determined from the analysis of a number of ancient metagenomic samples". Could provide a bit more detail here? What were the thresholds? What and how many samples did you use and how exactly did you determine the thresholds?

9) You start the discussion with what reads more like an introduction and you wanted to save space to make space for a more in depth discussion of e.g. the effects of database size, I would suggest you cut or condense the first two paragraphs of the discussion or move some of it to the introduction. That way I think the paper will be sharper.

10) On page 19, you conclude that "We believe that the features of aMeta ... make this workflow stand out in terms of accuracy and resource usage compared to other alternative analytical frameworks in the field." It's just a personal preference and I actually agree with you, but I think it might be better if you let the paper speak for itself!

11) I think it would be helpful if you could add some references for the denovo assembly of microbial genomes (page 20).

12) Again just personal preference, but you chose to end the paper by saying that "aMeta may currently not be as fast as HOPS when extensive multi-threading is available" but that you are working on several optimization schemes that will improve the speed of aMeta. While I think it's great to hear that you will continue to develop aMeta I think it's just not a very strong point to end on. So perhaps consider rephrasing that part or come up with a couple of sentences that sum up the strong points of your workflow?

Authors Response

Point-by-point responses to the reviewers' comments:

Reviewer 1

- 1) I think that it would be a good idea to add more explicit remarks and examples of the performance of aMeta in function of the number of microbial reads detected. Figure 3 and 5 provide some hints about it, but the real example shown in Figure 4 corresponds to a rather atypical case (Gok2) in which almost the whole genome of the pathogen was covered in the sequencing data. How will this look like, for instance, for the Gok4 individual that contains only ~2000 mapping reads (compared to the ~200K reads in Gok2)?

Authors' comment: to provide more information on performance of aMeta in function of the number of microbial reads detected, we included an additional Supplementary Figure 11 that compares counts of microbial reads detected by aMeta and HOPS with respect to the simulated ground truth. The horizontal dashed line in Supplementary Figure 11 marks a reasonable detection threshold of ~100-300 reads that one can apply to filter an abundance matrix. As it can be observed in the figure, lowering this detection threshold would deteriorate HOPS performance due to its high "dropout" (high false-positive and false-negative counts, i.e. the clouds of points grouping along the axes), therefore lowering the read number threshold brings too many false-positive hits. In contrast, the detection accuracy of aMeta can even slightly improve at read numbers as low as ~10-50. However, it is important to keep in mind that ~10-50 detected reads would be problematic to authenticate with e.g. mapDamage since a stable deamination profile can only be computed on a number of reads, at least ~200 to our experience. Therefore, we generally recommend aMeta users to keep filtering within range of ~100-300 reads which provides not only conservative detection but also possibility for reliable authentication analysis.

In addition, we explore aMeta performance with respect to the number of detected microbial reads in Figure 6 (Figure 5 in the previous version of manuscript) and Supplementary Figure 10 using a number of metrics such as Jaccard similarity, F1 score, detection accuracy and numbers of false-positive and false-negative counts. Here we come to a very similar conclusion: aMeta is consistently more accurate than HOPS in a wide range numbers of detected reads, and in theory can even be accurate for detecting very low-abundant microbes with only ~10-50 detected reads. However, we do not recommend this low detection limit due to potential authentication problems. We added the discussion about performance of aMeta in function of the number of reads detected in the main text, see lines 399-415, 705-719 and 360-376.

We also included a new Supplementary Figure 3 which demonstrates authentication output from aMeta for *Yersinia pestis* found in Gökhem 4 (Gok4) individual, N. Rascovan et al., Cell 2018. aMeta was successful in detecting and authenticating *Y.pestis* despite the lower amount of *Y.pestis* DNA in Gok4 individual. Note that in order to reliably detect and authenticate *Y.pestis* in Gok4 individual for Supplementary Figure 3, we had to merge 4 libraries sequenced in our lab, which resulted in ~49 mln reads in total, while for the example of Gok2 in Figure 4, only one library of ~12 mln reads was used.

- 2) The authors explored 18 ancient microbial species, but I believe that some additional pathogens that have been already successfully recovered in ancient DNA studies could be also included and tested too. I have then a following question regarding the detection and authentication capacity of aMeta for all these different ancient species when the number of target reads is limiting? (i.e., when the number of hits is closer to the 300 reads & 1000 k-mers threshold used by aMeta). *Y. pestis* is easy to detect and authenticate, but what about the other microbial species?

Authors' comment: to address this, we have performed two additional pathogen focused analyses. First, we simulated a new pathogen-enriched dataset with 5 pathogenic bacteria, 3 viruses and 1 eukaryotic pathogen, reviewed in Spyrou et al. Nat. Rev. Genet. 2019. Second, we evaluated aMeta on 36 ancient metagenomic libraries from 4 shotgun metagenomic studies that previously reported pathogens (Maixner et al. 2016, Warinner et al. 2014, Guallil et al. 2018 and Kay et al. 2014). The additional analyses were summarized in a new sub-section "Results - Replication on pathogen-enriched simulated and real datasets", see lines 540-599, Supplementary Figures 18-24 and Supplementary Table 1. We demonstrate that aMeta is more robust in discovering ancient pathogens compared to HOPS on simulated dataset, and default settings of aMeta are sufficient for confirmation of pathogen presence in real ancient metagenomic studies. Importantly, we addressed the limit of low-abundant pathogens and low coverage ancient metagenomic data. We demonstrated using the pathogen focused simulated dataset that aMeta with default settings was able to detect and authenticate most of the pathogens (41 out of 50) when sequencing depth was as low as 100 000 – 300 000 reads, Supplementary Figure 23. On real ancient shotgun metagenomic data, aMeta was also broadly successful in detecting pathogens (34 correct predictions out of 36) in libraries of varying sequencing depth

with the lowest at ~8 mln sequenced reads, but as simulations suggest, it can likely be even lower, Supplementary Figure 24 and Supplementary Table 1.

- 3) I think this detection and authentication threshold is very important to report more clearly because in the majority of cases the conservation of ancient pathogen genomes is very poor, so screenings must be performed on a very limited number of target reads, and decisions must be made on this limiting information to move into further sequencing or targeted enrichment strategies.

Authors' comment: we completely agree and have now added a paragraph into the Discussion section that provides detailed information about the detection and authentication thresholds used by aMeta, as well as motivation and intuition for using particular default values, see lines 684-719 and 364-370. We also demonstrate using simulated, Supplementary Figure 11, and real data, Supplementary Figure 24, that aMeta provides satisfactory accuracy of ancient microbiome profiling in a wide range of depth and breadth of coverage filters. Regarding sensitivity of aMeta's settings toward very limited number of target reads, please see the response to the previous question, as well as lines 573-599, where we provide justification based on simulated and read data that default settings of aMeta demonstrate satisfactory accuracy of detection and authentication even on a very limited number of target reads.

- 4) One of the main strengths of aMeta is the use of a dynamically generated database based on a subset of selected species. These species are selected based on seven validation parameters such as breadth of coverage and postmortem damage. However, during the benchmarking process, aMeta reported a slightly higher number of false positive hits, 14, compared to HOPS, 11 (supp fig 11). My main concern regarding this result, is that even after applying the filters proposed by the authors, HOPS, which doesn't use these filters, has a higher sensitivity for this type of errors. How do authors explain this?

Authors' comment: we thank the reviewer for pointing out the slightly higher false positive rate of aMeta compared to HOPS. After careful investigation, we found a minor error in the script used for producing the confusion matrix for aMeta detection which resulted in slight overestimation of aMeta's false-positive rate. The correct false-positive counts for aMeta are 9 compared to 12 for HOPS, which is demonstrated in the updated confusion matrix presented in Supplementary Figure 9 (Supplementary Figure 11 in the previous version of manuscript). We also modified

and extended the Supplementary Figure 10 (previously Supplementary Figure 12) which now explicitly shows both false-positive and false-negative counts of aMeta to be consistently lower than the ones for HOPS, as well as the accuracy of aMeta to be consistently higher than HOPS accuracy, in the wide range of the read number thresholds varying from 0 to 800 assigned reads. In addition, we updated Figure 6 (Figure 5 in the previous version) and Supplementary Figures 7 and 8 (previously Supplementary Figures 9 and 10), and modified the corresponding parts of the main text, see lines 345-415, and supplementary text that provide the details of detection error analysis. In summary, the additional robust filtering by breadth of coverage implemented in aMeta and its computational capacity to utilize a larger reference database result in both lower false-positive and lower false-negative detection error of aMeta compared to HOPS.

- 5) The authors mention that the k-mers unique to each taxa could be considered equivalent to the breadth of coverage information. I think it will be nice if they could develop this concept a bit further in the manuscript. For instance, how it conceptually relates to the breadth of coverage that can be calculated by SAMtools from a BAM file? Also, if I understand well, regions that are conserved among taxa are thus not counted because they are not unique to each taxa, right? Do the authors believe that masking such non-unique k-mers could have come confounding effects?

Authors' comment: to demonstrate that the number of unique k-mers reported by KrakenUniq could be considered equivalent to the breadth of coverage reported by Samtools, we produced a scatter plot of relation between these two metrics computed on the simulated ancient metagenomic dataset, see Supplementary Figure 1. The plot shows statistically significant correlation between the numbers of unique k-mers reported by KrakenUniq and breadth of coverage information computed by Samtools. We also added textual explanation for the relation between the two concepts to the Results, lines 128-138, and Discussion, lines 690-694, sections.

Regarding ignoring conserved regions and potential confounding effects, we believe that conserved regions can not unambiguously contribute to the depth and breadth of coverage information, therefore, indeed, one can imagine a hypothetical situation where abundance of some taxon is underestimated simply because the lack of "uniqueness" in the taxon's reference genome, e.g. when the reference genome of the taxon represents a mosaic of other taxa reference genomes. This would imply

that genome “uniqueness” can confound the abundance quantification. This problem, however, can be solved by considering longer reads and longer k-mers, which effectively increases the specificity of the analysis. That is, a longer read is less likely to map with the same affinity to multiple positions, and therefore will not be ignored when computing abundance and coverage. Similarly, in terms of k-mer-based analysis, longer k-mers tend to be more specific (unique) to a taxon and therefore more likely to contribute to the abundance and coverage information. In aMeta we select reads at least 31 bp long, and use KrakenUniq databases built with k-mer size $k = 31$, which is widely accepted to provide sufficient specificity of taxonomic assignment across the tree of life. We believe that long enough read length and k-mer size minimize the potential problem of confounding abundance and coverage by reference genome “uniqueness”. Also, in our opinion, an alternative attempt to account for non-unique k-mers that come from conserved regions may potentially result in even higher bias and stronger confounding effects.

Some additional Minor comments:

- 6) Introduction: The use of the term "endogenous" microbial communities. Does the definition of "endogenous" work well with the context in which it is used in the text? I've seen other terms, such as "targeted", or "of interest", or "host-associated", I would just double check that the term endogenous is ok here.

Authors' comment: we thank the reviewer for pointing this out, it has been edited in the new version of the manuscript, and the term "endogenous" has been replaced with "host-associated".

- 7) Intro, page 4, line35: I would not say that authentication error is specific of ancient metagenomics. This can be also an issue with modern samples, e.g., when detecting pathogens in clinical samples.

Authors' comment: thank you for the comment, we have removed the sentence on specificity of authentication error to ancient metagenomics, and rephrased the paragraph, see lines 85-95.

- 8) Supplementary information Page 3, line 8; Suppl Fig 2. The authors show that the Jaccard similarity reached a plateau starting with the Microbial NCBI NT database comprising around 110 billion characters. As one normally considers the numbers of genomes at the moment of creating a database instead of the number of characters, I think it would be nice

to provide information regarding the number of genomes from each type were included in this database: bacteria, viruses, archaea, fungi, protozoa and parasitic worms.

Authors' comment: we have modified the Figure 8 (that was previously Supplementary Figure 2, and was moved to the main text following another reviewer's suggestion) and explicitly included the numbers of reference genomes used for constructing each database. We also added exact numbers of microbial reference genomes (bacteria, viruses, archaea, fungi, protozoa and parasitic worms) used for constructing each database in the "Results – Effect of database size" subsection, lines 464-476.

Although, initially, we, indeed, calculated this figure in terms of numbers of genomes, we eventually decided that the number of characters might be a better representation of a "database size" when the range of genomic sizes varies greatly across databases. In fact, our smallest NCBI RefSeq database of complete reference genomes contains much fewer, i.e. ~1000 times fewer, genomes than our largest NCBI NT database. However, the number of characters in the latter is only ~4 times greater than in the former. This is because only well curated long (complete) reference genomes are included in the NCBI RefSeq database. In contrast, the NCBI NT, which is the GenBank database, has a lot of short (partially sequenced) genomes, i.e. only pieces of full genomes can be included. This bias might lead to an overestimation of "informativity" of NCBI NT and artificial inflation of its "database size", which we wanted to avoid. Nevertheless, we fully agree that both numbers, i.e. characters and reference genomes, should be provided, therefore we modified the text, lines 464-476, and the Figure 8 accordingly.

- 9) Supplementary information Page 4, line 11; Suppl Fig 3. When analyzing the impact of the database size, the authors observed that concatenating the hg38 human genome did not have an impact in the number of misaligned reads. As they mentioned in the manuscript, it is expected that with bigger databases the number of misaligned reads would get closer to zero. If possible, it would be interesting to see if this holds true after removing the hg38 genome and increasing the number of microbial genomes from the Microbial NCBI NT, keeping it in a reasonable amount of genomes to be able to run with the same amount of resources.

Authors' comment: we have now included Supplementary Figure 15 which replicates the decreasing profile for the number of (misaligned) reads mapped uniquely to

Y.pestis by sampling random bacteria from the Microbial NCBI NT database, this time without hg38 human reference genome, and up to 117 000 random reference genomes. We observed not only very similar qualitative behavior, i.e. bigger databases result in lower numbers of misaligned reads, but also quantitatively we obtained very similar, however slightly higher, numbers as previously reported for sampling random bacteria from NCBI RefSeq database, Figure 9 (previously Supplementary Figure 3, which was moved to the main text following another reviewer's suggestion). We assume that the slightly higher counts of misaligned reads observed for Microbial NCBI NT compared to NCBI RefSeq are related to the difference in the quality of reference genomes in the two databases, i.e. the same number but better quality reference genomes from NCBI RefSeq can "attract" more non-*Yersinia* reads, and thus result in fewer misaligned reads to *Y.pestis* reference genome. We also added a paragraph explaining the replication of misalignment to *Y.pestis* for the case of Microbial NCBI NT, lines 523-534.

10) Page 12, first line - Typo ".." Instead of "."

Authors' comment: thank you, it has been corrected.

11) Figure 3. Use of "kmers" instead of "k-mers" as in the rest of the manuscript.

Author's comment: thank you, the Figure 3 label and legend have been corrected.

Reviewer 2

1) The authors opted to go for KrakenUniq over other k-mer based classifiers and while I agree with that choice (as it has several advantages over say Kraken2) it might be worth pointing out what those advantages are.

Authors' comment: we decided to implement KrakenUniq within aMeta out of other k-mer based taxonomic classifiers because of two reasons. First, KrakenUniq delivers the number of unique k-mers metric, which makes microbial identification much more robust and represents a good approximation for breadth of coverage, see the new Supplementary Figure 1, which was added to the manuscript, we also added textual explanation, see lines 128-138 and 690-694. Second, the latest development of KrakenUniq [22] can be run in low-memory computational environments, which is a great advantage for the field of metagenomics, where reference databases typically comprise hundreds of thousands of reference genomes which are challenging to fit into computer memory. We discuss the

memory advantages of KrakenUniq in depth in lines 632-647 and Supplementary Figure 16.

2) Similarly, I would be interested to know why the authors opted to go for MALT over other mappers (e.g. minimap2 etc)?

Authors' comment: The main advantage of MALT and motivation for us to use it, was that MALT is a metagenomic-specific aligner which applies the Lowest Common Ancestor (LCA) algorithm. To the best of our knowledge, other traditional genomic aligners such as Bowtie2, BWA, minimap2 etc. do not perform LCA. The LCA algorithm is particularly important when working with heterogeneous metagenomic sequencing data. More specifically, when performing competitive mapping to multiple reference genomes, it is important to correctly handle the reads mapping with the same affinity to several references (multi-mapping reads). Traditional genomic aligners listed above would disregard the multi-mapping reads as ambiguous and non-informative. In contrast, the LCA algorithm keeps the multi-mapping reads within the taxonomic tree of related organisms and assigns the reads to the lower ancestor node in the tree. For example, if a read maps with the same number of mismatches to two species, the read will be assigned to their common genus and kept for the downstream analysis. We acknowledge that we did not explain the motivation well in the first version, and now a new paragraph with more explanation on MALT vs. other aligners has been added to the "Results" section of the manuscript, see lines 199-223.

3) A very minor point but on page 6 of the manuscript, the authors discuss why detecting microbial organisms solely based on "depth of coverage (or simply coverage)" might lead to false positive assignments. While I entirely agree with this point, it might be good to explain what exactly is meant by "coverage" (given the amount of confusion that exists in the field around terms like depth and breadth of coverage etc.). Simply saying coverage, might be unhelpful, I think. For clarity's sake, you could also introduce the term "evenness of coverage" in that same paragraph.

Authors' comment: We agree that the concept of evenness of coverage should be introduced in the same paragraph where we discuss the difference between depth and breadth of coverage in Figure 2. We have now modified this paragraph in the Results section, made explicit introduction of the concepts (depth, breadth and evenness of coverage), and extended the explanation of their difference, see lines 140-159. We hope that the distinction between the concepts is more clear now.

4) On page 7, you briefly discuss the effects of DB size on the rates of false assignments. I think

this is a very important point to make and I wonder if it might be worth discussing some of the results in more detail rather than banning them to the supplemental information.

Authors' comment: thank you for the suggestion, we share the view that the effect of database size is a very important message of the manuscript. The sub-section "Effect of database size" has now been moved from Supplementary to the Results in the main text, lines 449-538. In addition, two figures, i.e. Figure 8 and Figure 9, which provide in-depth explanation of why limited size databases can bias microbial discovery, have been moved from Supplementary to the Results in the main text.

5) Again a very minor point, but on page 8 you write that "KrakenUniq ... cannot control the authentication error". I know what you mean but I think it could perhaps be phrased a bit more clearly, so maybe rephrase or perhaps just drop "cannot control the authentication error because"?

Authors' comment: thank you for the suggestion, the phrase "cannot control the authentication error because" has now been removed.

6) On page 9, you write: "that would otherwise be technically impossible for MALT to build". I think maybe "handle" might be a better word here?

Authors' comment: we fully agree, the word "build" has been replaced with "handle", see line 233.

7) On page 11, you list the seven metrics you use to validate a true ancient microbe but what about others like average nucleotide identity (ANI)?

Authors' comment: thank you for this important suggestion, we have now integrated the ANI metric into the set of quality filters (this is now the 8th metric) used by aMeta for authentication analysis. We have also updated Figure 4 and Supplementary Figure 14 (Supplementary Figure 15 in the previous version of manuscript), which now report the average nucleotide identity (ANI) in addition to the barplot showing the numbers of reads mapped to the reference with a certain percent identity. We also added the explanation of how ANI is computed to the Supplementary Information S5 explaining aMeta's authentication scoring system. Finally, we recomputed aMeta and HOPS authentication scores for the simulated benchmark dataset, now taking the additional ANI metric into account, and updated the ROC-curve comparison, see Figure 7 (previously Figure 6), which, however, demonstrated very minor changes compared to the previous version without ANI. Finally, we introduced ANI in the main text see lines 245, 283 and 678.

8) On page 13, you write that the KrakenUniq and HOPS microbial abundance matrices were filtered using different thresholds for the number of assigned reads and later on in the same paragraph you write that "the default filtering thresholds were empirically determined from the analysis of a number of ancient metagenomic samples". Could provide a bit more detail here? What were the thresholds? What and how many samples did you use and how exactly did you determine the thresholds?

Authors' comment: we agree that this information was not stated clearly, and now we mention aMeta's defaults a few times throughout the manuscript, see e.g. lines 134, 163, 462, 588. We also added very detailed information about default aMeta filtering thresholds to the Discussion section, see lines 684-719 and 364-370, which provides motivation and intuition for using particular defaults in aMeta. We also demonstrate using simulated (see newly added Supplementary Figure 11), and real data (added Supplementary Figure 24) from 4 ancient shotgun metagenomic studies, that aMeta provides satisfactory accuracy of ancient microbiome profiling in a wide range of depth and breadth of coverage filters. We also additionally addressed the sensitivity of aMeta's settings toward filtering with respect to the number of detected target reads, see lines 573-599 and Supplementary Figure 23, where we provide justification based on simulated and read data that default settings of aMeta demonstrate satisfactory accuracy of detection and authentication even on a very limited number of target reads in e.g. shallow sequencing experiments.

9) You start the discussion with what reads more like an introduction and you wanted to save space to make space for a more in depth discussion of e.g. the effects of database size, I would suggest you cut or condense the first two paragraphs of the discussion or move some of it to the introduction. That way I think the paper will be sharper.

Authors' comment: thank you for the suggestion, the first two paragraphs in the Discussion section were edited and shortened, see lines 602-618.

10) On page 19, you conclude that "We believe that the features of aMeta ... make this workflow stand out in terms of accuracy and resource usage compared to other alternative analytical frameworks in the field." It's just a personal preference and I actually agree with you, but I think it might be better if you let the paper speak for itself!

Authors' comment: thank you, we agree and we removed this paragraph.

11) I think it would be helpful if you could add some references for the denovo assembly of microbial genomes (page 20).

Authors' comment: the references to de-novo assembly applied to modern, [38-41], and ancient metagenomic, [42], studies have been added, see lines 743-744.

12) Again just personal preference, but you chose to end the paper by saying that "aMeta may currently not be as fast as HOPS when extensive multi-threading is available" but that you are working on several optimization schemes that will improve the speed of aMeta. While I think it's great to hear that you will continue to develop aMeta I think it's just not a very strong point to end on. So perhaps consider rephrasing that part or come up with a couple of sentences that sum up the strong points of your workflow?

Authors' comment: thank you for your suggestion, we rephrased the paragraph to sound more positive and emphasize the strong sides of aMeta, see lines 783-797.

Second round of review

Reviewer 1

The corrected version of the manuscript addresses the concerns described during the first round of revision and pertinent changes have been made, greatly improving the manuscript. I appreciate the detailed explanation from the authors, as well as the new additions made in the manuscript and the software itself.

Here I outline my remaining questions and suggestions raised while reading the current version of the manuscript.

1. When the authors compare the specificity and sensitivity of aMeta and HOPS, they show a comparable number of false positive discoveries, 9 vs 12, respectively, and a higher rate of false negatives in HOPS (96) when compared with aMeta (60). However, it caught my attention that most of the false positive (8) were associated to a single sample, Sample 1, which is also the sample with the highest number of false negatives. As all datasets were simulated in the same way, I would expect the false positives to be equally distributed across all samples. Do you have an idea of what could cause this behavior?

2. Supplementary information, line 153. It is stated that the abundance matrix was binarized, where 0 corresponds to present, and 1 is absent, which is the opposite to what is presented in Supplementary Figures 6-8.

I believe that the revised version of the manuscript has improved the overall clarity of the article. The incorporation of the new section "Effect of database size" to the main text, helps to highlight the impact of using different databases. Moreover, this addition will likely serve users as guide when create their own databases.

Finally, I would like to also mention that aMeta was successfully tested by our team, thus proving that the software is ready to be used by new users. We also validated results that we

have obtained by other methods, on real data, thus also proving that the software serves the purposes for which it was created.

I have no additional comments to what I have mentioned above. Good job! I am sure aMeta will be widely adopted.

Authors' response:

Reviewer #1: The corrected version of the manuscript addresses the concerns described during the first round of revision and pertinent changes have been made, greatly improving the manuscript. I appreciate the detailed explanation from the authors, as well as the new additions made in the manuscript and the software itself.

Here I outline my remaining questions and suggestions raised while reading the current version of the manuscript.

1. When the authors compare the specificity and sensitivity of aMeta and HOPS, they show a comparable number of false positive discoveries, 9 vs 12, respectively, and a higher rate of false negatives in HOPS (96) when compared with aMeta (60). However, it caught my attention that most of the false-positives (8) were associated with a single sample, Sample 1, which is also the sample with the highest number of false negatives. As all datasets were simulated in the same way, I would expect the false positives to be equally distributed across all samples. Do you have an idea of what could cause this behavior?

Authors' comment: thank you for noticing this, it came as a surprise to us since we did not intend to simulate Sample 1 differently in any way from the rest of the samples. In fact, all the samples were simulated and processed simultaneously, and the gargammel code is available at https://github.com/NikolayOskolkov/aMeta/blob/main/gargammel_sim.sh). However, we confirm that the Sample 1 does seem to stand out in terms of elevated false-positive and false-negative counts obtained by both aMeta (8 false-positives out of 9 total, and 11 false-negatives out of 60 total) and HOPS (5 false-positives out of 12 total, and 12 false-negatives out of 96 total). After careful investigation, we believe there are a few reasons why this is the case. The overarching reason, in our opinion, is "the curse of small numbers", i.e. when dealing with only 10 samples and 35 microbes the effects of outliers may become profound, and adding or removing one sample or microbe can influence summary statistics of the dataset. Further, since abundances of microbial species were simulated randomly per sample (and only constrained by the total library size), it is possible that the Sample 1 happened by chance to be enriched by a combination of particular "hard case" microbes whose reference genomes might be contaminated. More specifically, *Mycobacterium avium* and *Rhodococcus hoagii* were false-positive discoveries by both aMeta and HOPS for Sample 1, which was likely due to the presence of *Mycobacterium riyadhense* and *Mycolicibacterium aurum* that are closely related to *Mycobacterium avium* and *Rhodococcus hoagii*, and were simulated to be highly abundant in Sample 1, which probably led to a number of *Mycobacterium riyadhense* and *Mycolicibacterium aurum* reads miss-assigned to *Mycobacterium avium* and *Rhodococcus hoagii*. Similarly, a possible explanation for a few other falsely detected microbes such as *Burkholderia mallei*, *Pseudomonas thivervalensis* and

Pseudomonas psychrophila is the presence of closely related *Ralstonia solanacearum* in the Sample 1. In addition, due to stochastic reasons, Sample 1 happened to have (together with Sample 6) the lowest median fraction of simulated reads across the microbial species, please see the figure below. Low-abundant microbes typically have a high error rate associated with their detection. We believe that part of the high error associated with Sample 1 may be explained by its enrichment by low-abundant microbial species.

2. Supplementary information, line 153. It is stated that the abundance matrix was binarized, where 0 corresponds to present, and 1 is absent, which is the opposite to what is presented in Supplementary Figures 6-8.

Authors' comment: thank you very much for noticing this, we have now corrected this in the supplementary information.

I believe that the revised version of the manuscript has improved the overall clarity of the article. The incorporation of the new section "Effect of database size" to the main text, helps to highlight the impact of using different databases.

Moreover, this addition will likely serve users as a guide when creating their own databases.

Finally, I would like to also mention that aMeta was successfully tested by our team, thus proving that the software is ready to be used by new users. We also validated results that we have obtained by other methods, on real data, thus also proving that the software serves the purposes for which it was created.